# Genetic diversity and evolutionary convergence of cryptic SARS- CoV-2 lineages detected via wastewater sequencing

**Devon A. Gregory[1], Monica Trujillo[2], Clayton Rushford[1], Anna Flury[3], Sherin Kannoly[4], Kaung Myat San[4], Dustin T. Lyfoung[5], Roger W. Wiseman[5], Karen Bromert[6], Ming-Yi Zhou[6], Ellen Kesler[6], Nathan J. Bivens[6], Jay Hoskins[7], Chung-Ho Lin[8], David H. O'Connor[5], Chris Wieberg[9], Jeff Wenzel[10], Rose S. Kantor[11]\*, John J. Dennehy[3,4]\*, Marc C. Johnson[1]\***

1 Department of Molecular Microbiology and Immunology, University of Missouri-School of Medicine, Columbia, Missouri, United States of America, 2 Department of Biological Sciences and Geology, Queensborough Community College of The City University of New York, New York City, New York, United States of America, 3 Biology Doctoral Program, The Graduate Center of The City University of New York, New York City, New York, United States of America, 4 Biology Department, Queens College of The City University of New York, New York City, New York, United States of America, 5 Department of Pathology and Laboratory Medicine, University of Wisconsin-Madison, Madison, Wisconsin, United States of America, 6 Genomics Technology Core, University of Missouri, Columbia, Missouri, United States of America, 7 Environmental Compliance Division, Engineering Department, Metropolitan St. Louis Sewer District, St. Louis, Missouri, United States of America, 8 Center of Agroforestry, School of Natural Resources, University of Missouri, Columbia, Missouri, United States of America, 9 Water Protection Program, Missouri Department of Natural Resources, Jefferson City, Missouri, United States of America, 10 Bureau of Environmental Epidemiology, Division of Community and Public Health, Missouri Department of Health and Senior Services, Jefferson City, Missouri, United States of America, 11 Department of Civil and Environmental Engineering, University of California, Berkeley, California, United States of America

\* rkantor@berkeley.edu (RSK); John.Dennehy@qc.cuny.edu (JJD); marcjohnson@missouri.edu (MCJ)

**Data Availability Statement:** The MO raw sequence reads are available in NCBI's SRA under the BioProject accession PRJNA748354. The NY raw sequence reads are available in NCBI's SRA

## Abstract

Wastewater-based epidemiology (WBE) is an effective way of tracking the appearance and spread of SARS-COV-2 lineages through communities. Beginning in early 2021, we implemented a targeted approach to amplify and sequence the receptor binding domain (RBD) of SARS-COV-2 to characterize viral lineages present in sewersheds. Over the course of 2021, we reproducibly detected multiple SARS-COV-2 RBD lineages that have never been observed in patient samples in 9 sewersheds located in 3 states in the USA. These cryptic lineages contained between 4 to 24 amino acid substitutions in the RBD and were observed intermittently in the sewersheds in which they were found for as long as 14 months. Many of the amino acid substitutions in these lineages occurred at residues also mutated in the Omicron variant of concern (VOC), often with the same substitutions. One of the sewersheds contained a lineage that appeared to be derived from the Alpha VOC, but the majority of the lineages appeared to be derived from pre-VOC SARS-COV-2 lineages. Specifically, several of the cryptic lineages from New York City appeared to be derived from a common ancestor that most likely diverged in early 2020. While the source of these cryptic lineages has not been resolved, it seems increasingly likely that they were derived from long-term patient infections or animal reservoirs. Our findings demonstrate that SARS-COV-2 genetic diversity is greater than what is commonly observed through routine SARS-CoV-2 surveillance.

under the BioProject accession PRJNA715712. The indicated NCBI SRA data can be found at https://www.ncbi.nlm.nih.gov/sra. The script used for the haplotype condensation can be found at https://github.com/degregory/SARS2_Cryptic_WW/blob/main/Deconv_condenser.py.

**Funding:** This project has been funded in part with federal funds from the NIDA/NIH (www.nida.nih.gov/) under contract numbers 1U01DA053893-01 to JW and MCJ and by the New York City Department of Environmental Protection (www.nyc.gov/dep) under contract number 1484-RDOP to JJD. This work was supported by financial support through Rockefeller Regional Accelerator for Genomic Surveillance (www.rockefellerfoundation.org,133 AAJ4558), Wisconsin Department of Health Services Epidemiology and Laboratory Capacity funds (www.dhs.wisconsin.gov, 144 AAJ8216) to DHO. The work was supported by funds from the California Department of Health (www.dhcs.ca.gov/) to RSK. The funders had no role in study design, data collection and analysis, decision to publish, or preparation of the manuscript. The DEP played no role in study design, data collection, analysis or preparation of the manuscript. However, they did require that they review the manuscript and approve its publication.

**Competing interests:** The authors have declared that no competing interests exist.

Wastewater sampling may more fully capture SARS-CoV-2 genetic diversity than patient sampling and could reveal new VOCs before they emerge in the wider human population.

## Author summary

During the COVID-19 pandemic, wastewater-based epidemiology has become an effective public health tool. Because many infected individuals shed SARS-CoV-2 in feces, wastewater has been monitored to reveal infection trends in the sewersheds from which the samples were derived. Here we report novel SARS-CoV-2 lineages in wastewater samples obtained from 3 different states in the USA. These lineages appeared in specific sewersheds intermittently over periods of up to 14 months, but generally have not been detected beyond the sewersheds in which they were initially found. Many of these lineages may have diverged in early 2020. Although these lineages share considerable overlap with each other, they have never been observed in patients anywhere in the world. While the wastewater lineages have similarities with lineages observed in long-term infections of immunocompromised patients, animal reservoirs cannot be ruled out as a potential source.

## 1. Introduction

SARS-CoV-2 is shed in feces of infected individuals [1,2], and SARS-CoV-2 RNA can be extracted and quantified from community wastewater to provide estimates of SARS-CoV-2 community prevalence [3,4]. This approach is especially powerful since it randomly samples all community members and can detect viruses shed by individuals whose infections are not recorded, such as asymptomatic individuals, those who abstain from testing, or those who test at home [5,6]. Additionally, SARS-CoV-2 RNA isolated from wastewater can be sequenced using high-throughput sequencing technologies to define the composition of variants in the community [7–9].

The continuing evolution of SARS-CoV-2 [10] and the appearance of variants of concern (VOC), such as the Omicron VOC [11], highlight the importance of maintaining a vigilant watch for the emergence of unexpected, novel variants. The fact that the origins and early spread of the Alpha and Omicron VOCs were not observed strongly motivates efforts to detect and monitor novel variants [12]. However, whole genome sequencing of SARS-CoV-2 RNA isolated from wastewater often suffers from low sequencing depth of coverage in epidemiologically relevant areas of the genome, such as the Spike receptor binding domain (RBD) [13–15]. Additionally, because wastewater may contain a mixture of viral lineages and whole genome sequencing relies on sequencing small fragments of the genome, computational strategies to identify variants with linked mutations often fail to identify lineages present at low concentrations [16]. These features have made it difficult to detect unexpected, novel variants from wastewater samples from whole genome sequencing data.

To address these issues, we developed a "targeted" sequencing approach that amplifies and sequences the Spike RBD of the SARS-CoV-2 genome as a single amplicon (Fig 1A) [8,9]. Since the Spike RBD is relevant to SARS-CoV-2 infectivity, transmission, and antibody-mediated neutralization [17–21], this approach ensures that the RBD receives high sequencing coverage. Additionally, RBD sequencing enables linkage of polymorphisms, forming short, phased haplotypes [16]. These phased haplotypes permit easier lineage identification, even at

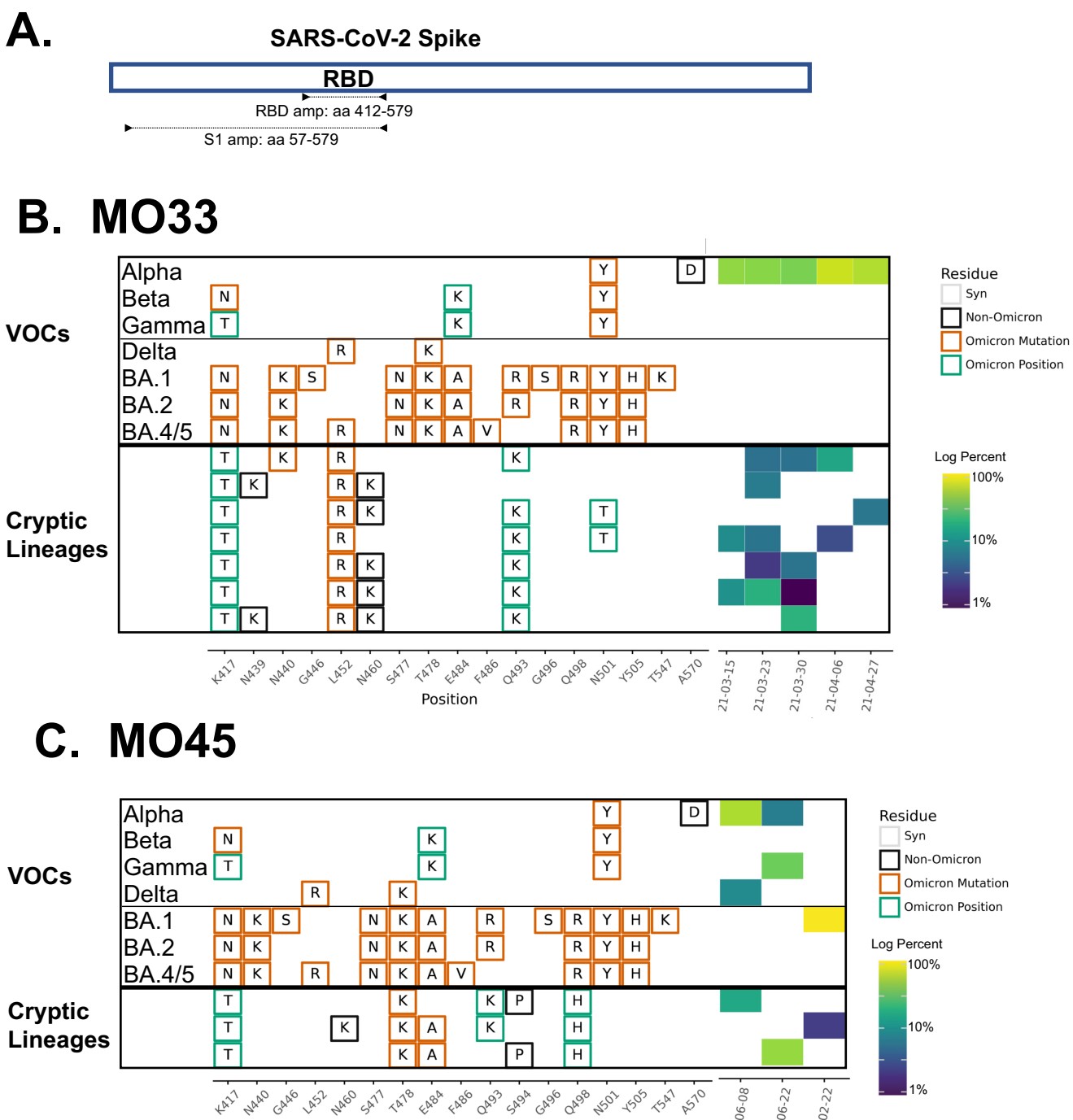

**Fig 1. RBD amplification.** A. Schematic of regions targeted by the RBD and S1 primer sets (see Methods for primer sequences). Overview of the SARS-COV-2 Spike RBD lineages identified in B. the MO33 sewershed and C. the MO45 sewershed. Each row represents a unique lineage and each column is an amino acid position in the Spike protein (left). Amino acid changes similar to (green boxes) or identical to (orange boxes) changes in Omicron (BA.1, BA.2 or BA.5) are indicated. Synonymous changes (syn) are indicated in gray. The major US VOCs (Alpha, Beta, Gamma, BA.1, BA.2, and BA.5) are indicated. The heatmap (right) illustrates lineage (row) detection by date (column), colored by the $\log_{10}$ percent relative abundance of that lineage. Uncondensed output in S1 and S2 Data.

low concentrations, if the targeted sequence (s) are rich in lineage-defining polymorphisms [9].

Using our targeted sequencing approach, we identified and previously reported circulating VOCs in different sewersheds around the United States [8,9]. Variant frequencies in these sewersheds closely tracked VOCs frequency estimates from clinical sampling in the same areas [8,9]. However, in some locations, we noted the presence of cryptic lineages not observed in clinical samples anywhere in the world. Several of these lineages contained amino acid substitutions that were rarely reported in global databases such as gisaid.org [22–24] (e.g., N460K, Q493K, Q498Y, and N501S) [8]. Interestingly, polymorphisms in these lineages show considerable overlap with the Omicron VOC and with each other, suggesting convergent evolution due to similar selective pressures.

Here we describe an expanded set of cryptic lineages from multiple locations around the United States. While each sewershed contains its own signature lineages and at least some of the lineages appear to have diverged independently from one another, we present evidence that some likely shared a common ancestor. Finally, we show evidence of strong positive selection and rapid divergence of these lineages from ancestral SARS-CoV-2.

## 2. Results

Beginning in early 2021, wastewater surveillance programs including RBD amplicon sequencing (Fig 1A) were independently implemented in Missouri [9] and NYC [25]. A similar strategy was subsequently adopted in California by the University of California, Berkeley wastewater monitoring laboratory (COVID-WEB). All of the sequence output was analyzed with our previously described SAM Refiner pipeline [9], which is designed to remove PCR-generated chimeric sequences. While the vast majority of sequences observed with this method matched to known lineages identified in patients, reproducible lineages that did not match the known circulating lineages were also detected. Herein, we refer to each RBD haplotype with a unique combination of amino acid changes as a *lineage*, and combinations of lineages that all have specific amino acid changes in common as *lineage classes*. Amino acid combinations identified that have not been seen previously from patients are referred to as *cryptic* lineages. Here we describe cryptic lineages detected from January 1, 2021 through March 15, 2022.

For display purposes, for most sewersheds (those with >3 cryptic lineage-positive samples) individual polymorphisms were only displayed if they were present in at least two independent samples. Further, individual lineages were only displayed if they were over 2% of the total signal in at least one sample, or were present in at least 2 independent samples. The detailed display criteria is outlined in Materials and Methods. The complete uncompressed data sets are included in S1–S9 Data.

### 2.1 Lineage persistence and evolution over time

In total, cryptic lineages were observed in 9 sewersheds across 3 states (Table 1) out of approximately 180 sewersheds that were routinely monitored. Each cryptic lineage class was generally unique to a sewershed. These lineages contained between 4–24 non-synonymous substitutions, insertions, and deletions. In some cases, lineages were detected for a short duration but with multiple similar co-occurring sequences. For example, in Missouri sewershed MO33, a lineage class containing 4–5 RBD amino acid changes were consistently detected at low relative abundances from March 15 to the end of April 2021 (Fig 1B and Table 1 and S1 Data). A total of 7 unique sequences were spread across the 5 sampling events in this date range, and multiple unique sequences co-occurred within a given sample.

Meanwhile, in other sewersheds, cryptic lineages were detected briefly, before disappearing, and then reappearing many months later. For example, in Missouri sewershed MO45, lineages

**Table 1. Overview of cryptic lineage detection.**

| Location | Date range when lineages appeared | Days within range | Number of samples | Number of RBD mutations |
|---|---|---|---|---|
| NY2 | 8/16/21-02/28/22 | 170 | 10 | 4–18 |
| NY3 | 1/31/21 [8] -3/14/22 | 437 | 7 | 16–24 |
| NY10 | 4/4/21-11/29/21 | 239 | 22 | 4–11 |
| NY11 | 4/19/21-11/22/22 | 217 | 20 | 4–9 |
| NY13 | 10/26/21-2/14/22 | 111 | 5 | 12–15 |
| NY14 | 5/10/21-10/18/21 | 161 | 9 | 8–15 |
| MO33 | 3/15/21-4/27/21 | 43 | 12 | 4–6 |
| MO45 | 6/8/21-2/22/22 | 259 | 3 | 4–5 |
| CA | 11/4/21-12/21/21 | 47 | 3 | 16 |

were first detected in June 2021 and then were not seen again until February 2022 (Fig 1C and Table 1 and S2 Data). The longest observed lineage class was in sewershed NY3 where we previously reported a lineage class from January 2021 [8] that was detected sporadically until March 2022 (Fig 2A and Table 1 and S3 Data). On average, cryptic lineages lasted for around 6 months, such as the lineage class from NY14 which lasted from May to October, 2021 (Fig 2B and Table 1 and S4 Data).

Each sewershed had its own unique set of lineages, but these lineages were not static. For instance, in NY10, the lineages first detected in April 2021 contained 4–5 RBD amino acid changes, but by October and November the lineages contained at least 6–8 RBD amino acid changes (Fig 3 and Table 1 and S5 Data).

In some cases, the sewersheds contained more than one lineage class. For instance, the NY11 sewershed contained several closely related lineages (class A) starting in April 2021, but a new lineage class (class B) was detected starting in August 2021. These two classes were clearly distinct with very few amino acid changes in common (Fig 4A and Table 1 and S6 Data).

Overall, specific lineage classes persisted within, but did not spread beyond, their individual sewersheds, with one notable exception. A cryptic lineage detected on August 16, 2021 in NYC sewershed NY2 precisely matched a lineage detected in sewershed NY11 between June-September 2021 (Fig 4A and 4B indicated by *, and S6 and S7 Data). The NY11 and NY2 sewersheds do not border each other, but are not separated by any bodies of water.

In addition to amino acid changes, several of the lineages observed in these sewersheds contained amino acid deletions near positions 445 and 484. For instance, lineages NY11 contained 445–446 deletions, NY14 contained 444–445 deletions, NY3 and NY11 contained a deletion at position 484, and NY2 contained a deletion at position 483 (Figs 2 and 4).

Most cryptic lineages detected did not contain changes consistent with being derived from any known VOCs. The one exception was a lineage class containing amino acid changes N501Y and A570D in NY13 that first appeared on September 26, 2021, which suggested possible derivation from the Alpha VOC (Fig 5 and Table 1 and S8 Data). The Alpha VOC had been the dominant lineage in NYC between April and June 2021, but by September 26, 2021, it had been supplanted by Delta VOC and was no longer being detected in NYC [26].

## 2.2 Rare and concerning amino acid changes are common in cryptic lineages and are sometimes shared with Omicron

In November 2021, the Omicron VOC was first detected in South Africa. This VOC contained eleven changes in the Spike protein between amino acids 410–510. Of these eleven amino acid changes, four (K417T, S477N, T478K, and N501Y) were present in previous VOCs. The

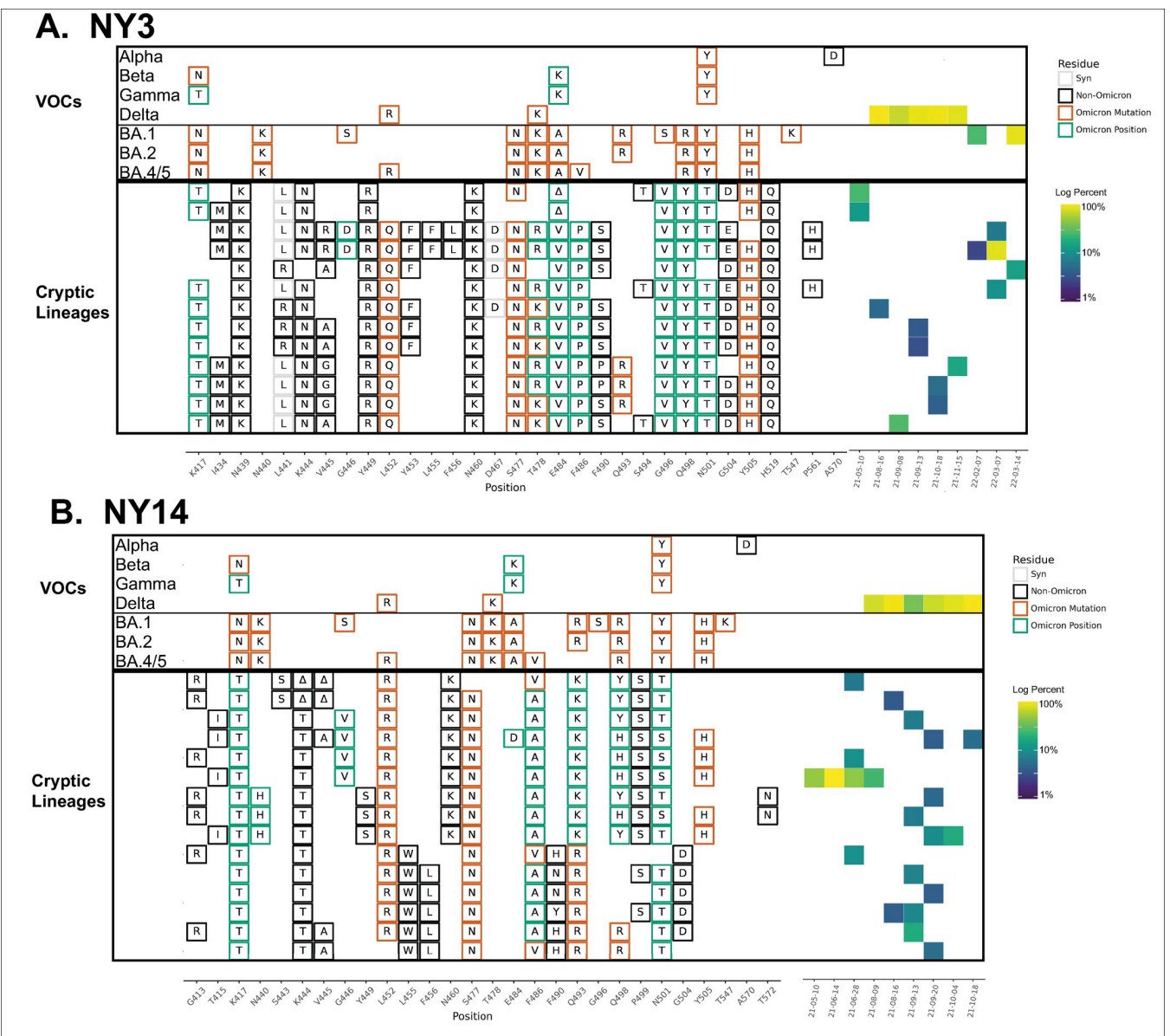

**Fig 2. NY3 and NY14 RBD amplifications.** Overview of the SARS-COV-2 Spike RBD lineages identified from the A. NY3 and B. NY14 sewersheds. Amino acid changes similar to (green boxes) or identical to (orange boxes) changes in Omicron (BA.1, BA.2 or BA.5) are indicated. Synonymous changes (syn) are indicated in gray. The major US VOCs (Alpha, Beta, Gamma, BA.1, BA.2, and BA.5) are indicated. The heatmap (right) illustrates lineage (row) detection by date (column), colored by the $\log_{10}$ percent relative abundance of that lineage. Uncondensed output in S3 and S4 Data.

remaining seven amino acid changes were rare prior to the Omicron VOC. All seven of these new amino acid changes had been detected in at least one of the wastewater lineages: N440K (MO33), G446S (NY2), E484A (MO45, NY10, NY11, NY2, NY13, CA), Q493R (NY3, NY14), G496S (NY2), Q498R (NY13, NY14), and Y505H (NY2, NY3, NY13, NY14, CA) (Figs 2 and 4–6, and S3 and S4 and S6–S9 Data). None of the wastewater lineages have combinations of amino acid changes consistent with having a common ancestor with Omicron and most were initially detected prior to the emergence of Omicron. However, these shared amino acid changes suggest that the cryptic lineages were under selective pressures similar to those that shaped the Omicron lineage.

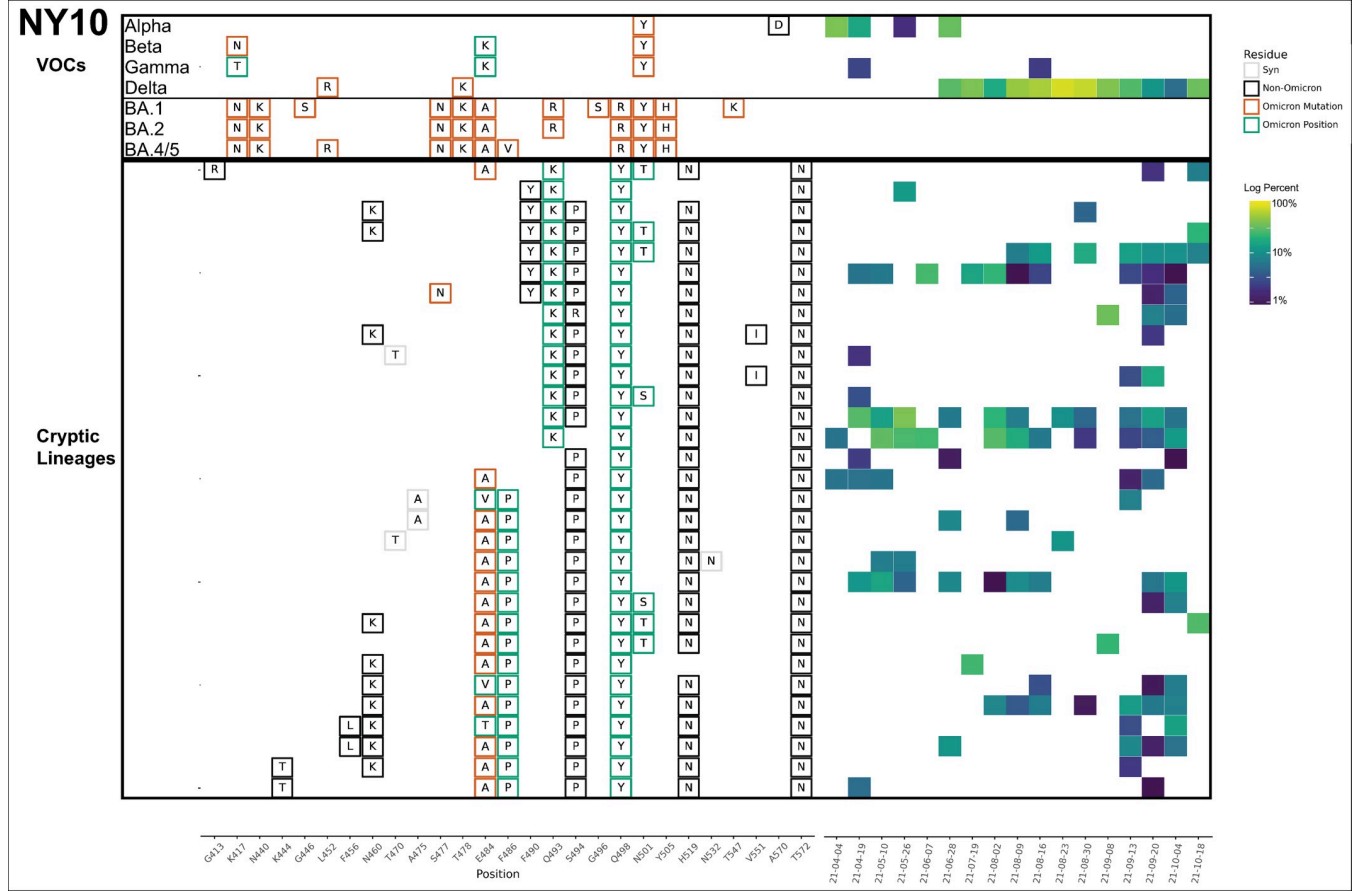

**Fig 3. NY10 RBD amplifications.** Overview of the SARS-COV-2 Spike RBD lineages identified from the NY10 sewershed. Amino acid changes similar to (green boxes) or identical to (orange boxes) changes in Omicron (BA.1, BA.2 or BA.5) are indicated. Synonymous changes (syn) are indicated in gray. The major VOCs during this time period (Alpha, Beta, Gamma, BA.1 and BA.2) are indicated. The heatmap (right) illustrates lineage (row) detection by date (column), colored by the $\log_{10}$ percent relative abundance of that lineage. Uncondensed output in S5 Data.

Although each sewershed with cryptic lineages had its own signature combinations of amino acid changes, many of these changes were recurring among multiple sewersheds. Some of the more striking examples are described below.

*N460K.* All nine of the sewersheds contained lineages with this change. Changes at this position are known to lead to evasion of class I neutralizing antibodies [27,28]. However, this amino acid change was very rare, appearing in less than 0.01% of sequences in GISAID [22–24] submitted by March 15, 2022 (S1 Table).

*K417T.* Eight of the nine sewersheds contained lineages with the amino acid change K417T. Changes at this position are common and are known to participate in evasion from class I neutralizing antibodies [27,28]. Although K417T was present in the Gamma VOC, K417N is the more common amino acid change at this position. The K417N amino acid change was not observed in any of the wastewater cryptic lineages.

*N501S/T.* The amino acid changes N501S and N501T were seen in four and seven of the nine sewersheds, respectively. Changes at this position directly affect receptor binding and can affect the binding of multiple classes of neutralizing antibodies [19,29,30]. Although mutations at this position are very common, the most common change by far is N501Y, which was present in multiple VOCs. By contrast, N501S and N501T were present in less than 0.01% and 0.1% of sequences in GISAID [22–24] submitted by March 15, 2022 (S1 Table).

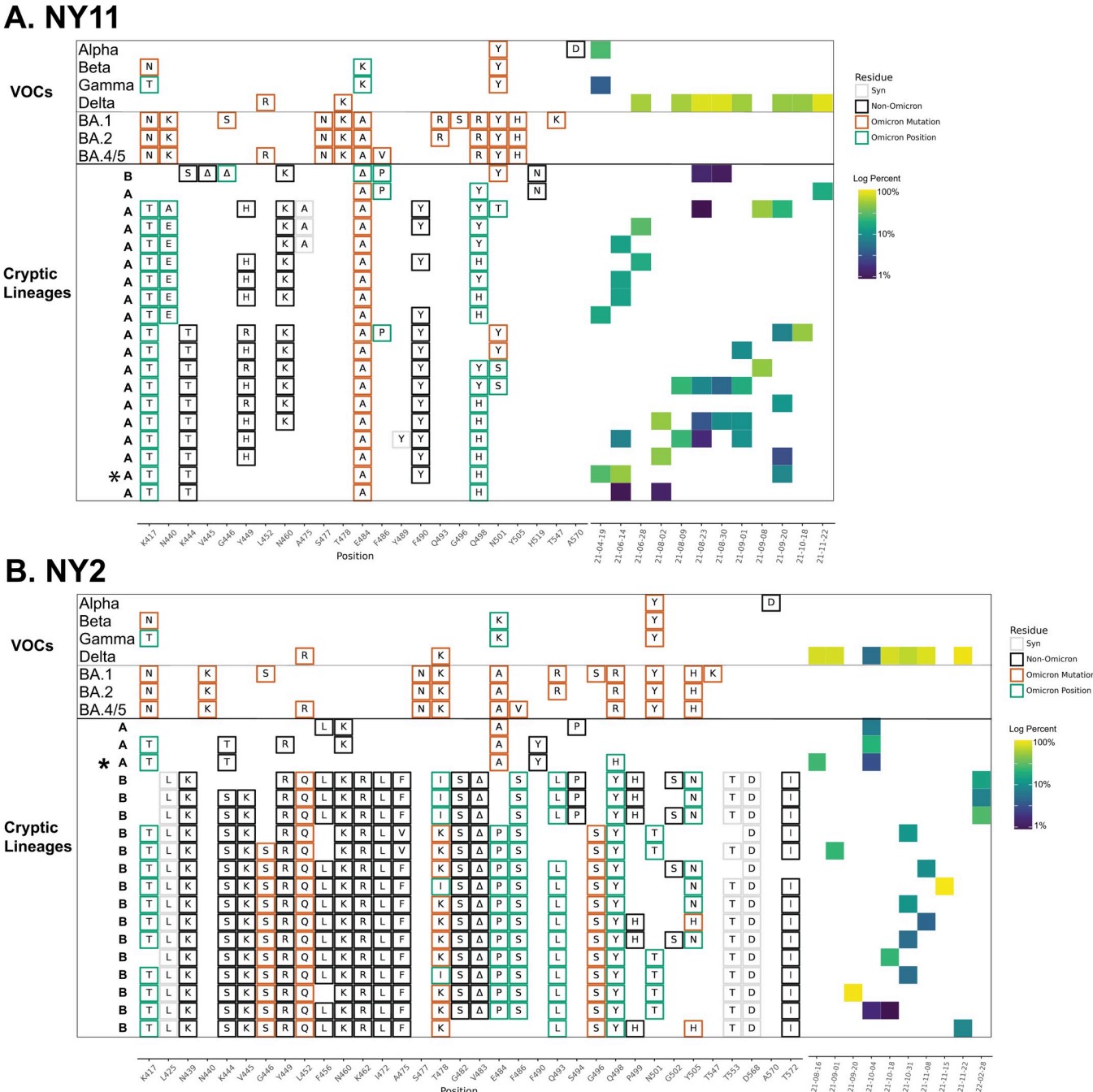

**Fig 4. NY11 and NY2 RBD amplifications.** Overview of the SARS-COV-2 Spike RBD lineages identified from the NY11 and NY2 sewershed. Lineages designated A and B belong to two lineages groups that appear unrelated. Amino acid changes similar to (green boxes) or identical to (orange boxes) changes in Omicron (BA.1, BA.2 or BA.5) are indicated. Synonymous changes (syn) are indicated in gray. The major VOCs during this time period (Alpha, Beta, Gamma, BA.1 and BA.2) are indicated. The heatmap (right) illustrates lineage (row) detection by date (column), colored by the $\log_{10}$ percent relative abundance of that lineage. Lineage detected in both sewersheds indicated with an asterisk. Uncondensed output in S6 and S7 Data.

*Q498H/Y.* Seven of the nine sewersheds in this study contained lineages with the amino acid change Q498H or Q498Y. It should be noted that Q498Y differs from the Wuhan ancestral sequence by two nucleotide substitutions at the 498th codon (CAA→TAC). Q498H (CAA→CAC) is a necessary intermediary in this transition as TAA encodes a stop codon. In

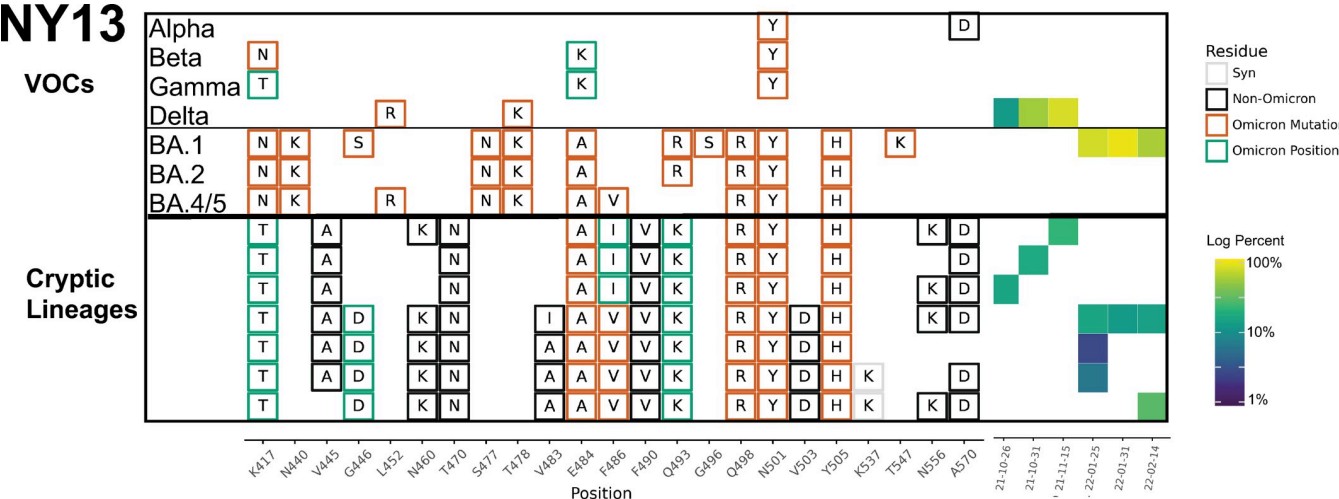

**Fig 5. NY13 RBD amplifications.** Overview of the SARS-COV-2 Spike RBD lineages identified from the NY13 sewershed. Amino acid changes similar to (green boxes) or identical to (orange boxes) changes in Omicron (BA.1, BA.2 or BA.5) are indicated. Synonymous changes (syn) are indicated in gray. The major VOCs during this time period (Alpha, Beta, Gamma, BA.1 and BA.2) are indicated. The heatmap (right) illustrates lineage (row) detection by date (column), colored by the log$_{10}$ percent relative abundance of that lineage. Uncondensed output in S8 Data.

several cases both Q498H and Q498Y were seen in association with particular lineage classes including in NY2, NY11, NY14 and CA (Figs 2B, 4 and 6). Changes at this position directly affect receptor binding [19,29,30]. Notably, Q498H and Q498Y have been associated with mouse adapted SARS-CoV-2 lineages [31–33]. Both of these amino acid changes are very rare, appearing in less than 0.01% of sequences in GISAID [22–24] submitted by March 15, 2022. Prior to November 2021, Q498Y had never been seen in a patient sample (S1 Table).

*E484A.* Six of the nine sewersheds contained lineages with the amino acid change E484A. Changes at this position are known to participate in evasion from class II neutralizing antibodies [27,28]. Prior to the emergence of Omicron in November 2021, E484A was present in about 0.01% of sequences submitted to GISAID [22–24] (S1 Table).

*Q493K.* Five of the nine sewersheds contained lineages with the amino acid change Q493K. Changes at this position directly affect receptor binding and can affect the binding of multiple

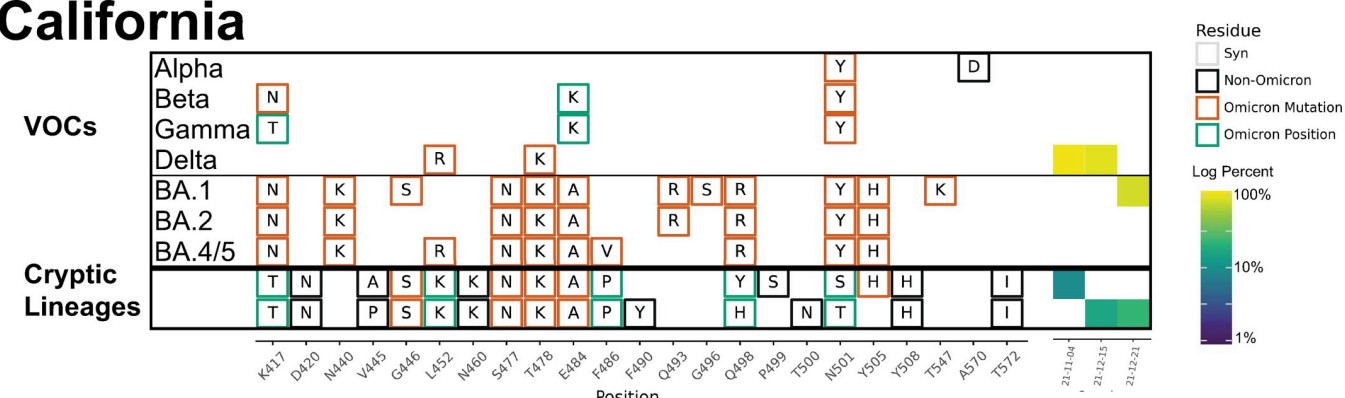

**Fig 6. Overview of the SARS-COV-2 Spike RBD lineages identified from the California sewershed.** Amino acid changes similar to (green boxes) or identical to (orange boxes) changes in Omicron (BA.1, BA.2 or BA.5) are indicated. Synonymous changes (syn) are indicated in gray. The major VOCs during this time period (Alpha, Beta, Gamma, BA.1 and BA.2) are indicated. The heatmap (right) illustrates lineage (row) detection by date (column), colored by the log$_{10}$ percent relative abundance of that lineage. Uncondensed output in S9 Data.

classes of neutralizing antibodies [19,27–30,34]. This amino acid change is biophysically very similar to the Q493R mutation in Omicron. However, the Q493K amino acid change was very rare in patient derived sequences, appearing in less than 0.01% of sequences in GISAID [22–24] submitted by March 15, 2022 (S1 Table).

*Y505H.* Five of the nine sewersheds contained lineages with the amino acid change Y505H. Prior to the emergence of Omicron in November 2021, Y505H was present in about 0.01% of sequences submitted to GISAID [22–24] (S1 Table).

*K444T and K445A.* The amino acid changes K444T and K445A were each seen in four of the nine sewersheds. Changes at these positions are known to participate in evasion from class III neutralizing antibodies [28]. However, these amino acid changes were very rare, each appearing in less than 0.01% of sequences in GISAID [22–24] submitted by March 15, 2022 (S1 Table).

*Y449R.* Three of the nine sewersheds contained lineages with the amino acid change Y449R. This change is noteworthy because, as of March 15, 2022, no sequences with this amino acid change had been submitted to GISAID [22–24] (S1 Table).

## 2.3 Long-read sequencing of S1 identifies substantial NTD modifications and suggests high dN/dS ratio

With each sample that contained novel cryptic lineages, attempts were made to amplify a larger fragment of the S1 domain of Spike. Amplification of larger fragments from wastewater is often inefficient, but sometimes can be achieved. To gain more information about the S1 domain of Spike and independently confirm the authenticity of the RBD lineages, we optimized a PCR strategy that amplifies 1.6 kb of the SARS-COV-2 Spike encompassing amino acids 57–579. These fragments were then either subcloned and sequenced or directly sequenced using Pacific Biosciences HiFi sequencing (Fig 7A).

The S1 amplification from the MO33 and MO45 sewersheds contained the RBD amino acid changes previously seen and each contained 3 additional amino acid changes upstream from the region sequenced using the targeted amplicon strategy described above (Fig 7A). Many of the S1 amplifications from the NY10, NY11, NY13 and NY14 sewersheds contained numerous changes in S1 (Fig 7A). In particular, many of the sequences contained deletions near amino acid positions 63–75, 144, and 245–248. All three of these areas are unstructured regions of the SARS-COV-2 spike where deletions have been commonly observed in sequences obtained from patients [35]. Two distinct S1 sequences were detected from the NY14 sample collected on June 28, 2021. Interestingly, the first sequence contained 13 amino acid changes which matched the RBD sequences from the same sewershed. The second sequence did not match any lineage that had been seen before, though it contained several mutations that were commonly seen in other cryptic lineages (see section 2.2). This second sequence presumably represented a unique lineage that had not been detected by routine wastewater surveillance.

A single S1 sequence was obtained from the NY13 samples collected on October 31, 2021. This sequence generally matched the RBD sequence from the same date, but did contain minor variations. Importantly, the S1 sequence contained deletions at positions 69–70 and 144, which, along with the amino acid changes N501Y and A570D, match the changes found in the Alpha VOC lineage. This information is consistent with the NY13 lineages being derived from the Alpha VOC.

Comparing the number of non-synonymous to synonymous mutations in a sequence can elucidate the strength of positive selection imposed on a sequence. The ratios of non-synonymous and synonymous mutations in this region of S1 from the Alpha, Delta, and Omicron

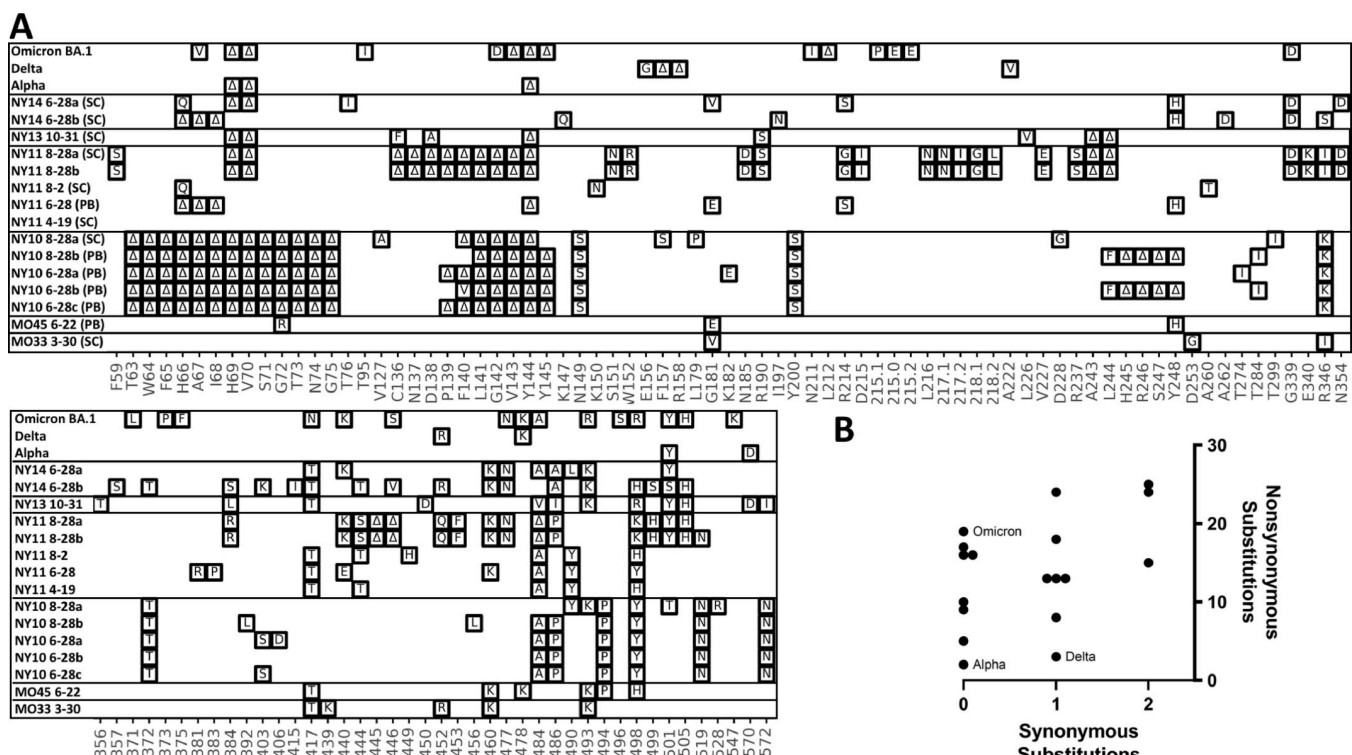

**Fig 7. S1 amplifications.** A. Overview of the SARS-COV-2 Spike S1 lineages in the Alpha, Delta, Omicron VOCs and six of the sewersheds with cryptic lineages. S1 amplifications were sequenced by subcloning (SC) and Sanger sequencing, or were sequenced using a PacBio (PB) deep sequencing. B. Plot of the number of synonymous and non-synonymous changes in the S1 sequences shown.

VOCs (BA.1) were 19/0, 2/0, and 4/1, respectively. It was not possible to calculate the formal dN/dS ratios since many of the sequences did not have synonymous mutations in this region, so instead the numbers of non-synonymous and synonymous mutations were plotted. The cryptic lineages contained 5 to 25 total non-synonymous mutations and 0 to 2 total synonymous mutations (Fig 7B).

## 2.4 Cryptic lineages from NCBI suggest an early common ancestor for many of the NYC lineages

In addition to RBD amplicon sequencing performed in our laboratories, we downloaded the 5609 SARS-CoV-2 wastewater fastq files from NCBI's Sequence Read Archive (SRA) that were publicly available on NCBI on January 21, 2022 (not including submissions from our own groups). We screened these sequences for cryptic lineages by searching for recurring amino acid changes seen via RBD amplicon sequencing (K444T, Y449R, N460K, E484A, F486P, Q493K, Q493R, Q498H, Q498Y, N501S, N501T, and Y505H) (see above and S1 Table), requiring at least two of these mutations with a depth of at least 4 reads. This strategy identified samples from 15 sewersheds (Table 2). Four were collected from unknown sewersheds in New Jersey and California in January 2021. The other 11 were collected by the company Biobot from NYC between June and August 2021. All but one of the lineages closely matched the cryptic sequences that had been observed via RBD amplicon sequencing from the same sewershed. The one exception was SRR16038150, which contained 4 amino acid changes that had not been seen in any of the previous sewershed samples in the same combination. The Biobot sequences were 40–96% complete and appeared to contain 30–100% cryptic lineages based on

**Table 2. Cryptic lineage whole genome sequences from nationwide surveys.**

| SRA Accession | State | Submitter | Sample Date | Percent cryptic lineage | Genome coverage | Sewershed | PANGO assignment | RBD Changes |
|---|---|---|---|---|---|---|---|---|
| SRR17120725 | CA | Aquavitas | 2021-01-04 | 7% | 27,403 | n/a[a] | ND[b] | E484A/Q498H/H519N |
| SRR16638981 | NJ | Aquavitas | 2021-01-18 | 7% | 28,185 | n/a[a] | ND[b] | E484A/Q498H/H519N |
| SRR16542155 | NJ | Aquavitas | 2021-01-18 | 7% | 27,295 | n/a[a] | ND[b] | E484A/Q498H/H519N |
| SRR16362183 | NJ | Aquavitas | 2021-01-04 | 100% | 15,217 | n/a[a] | ND[b] | E484A/Q498H/H519N |
| SRR16038150 | NY | Biobot Analytics | 2021-08-17 | 79% | 28,227 | NY2 | B.1.503 | Y449P/E484A/F490Y/Q498H |
| SRR16038156 | NY | Biobot Analytics | 2021-08-09 | 92% | 24,595 | NY11 | B.1.503 | K417T/K444T/Y449H/N460K/E484A/F490Y/Q498H |
| SRR15706711 | NY | Biobot Analytics | 2021-08-09 | 100% | 11,877 | NY11 | ND[b] | K417T/K444T/Y449H/N460K/E484A/F490Y/Q498H/ A570D |
| SRR15384049 | NY | Biobot Analytics | 2021-07-12 | 99% | 24,001 | NY10 | B.1 | Q493K/Q498Y/H519N/T572N) |
| SRR15291305 | NY | Biobot Analytics | 2021-07-05 | 100% | 22,316 | NY11 | P.1.15 | K417T/K444T/Y449H/E484A/F490Y/Q498H |
| SRR15291304 | NY | Biobot Analytics | 2021-07-04 | 100% | 28,634 | NY10 | B.1 | Q493K/Q498/H519N/T572N |
| SRR15202285 | NY | Biobot Analytics | 2021-06-28 | 100% | 12,209 | NY2 | ND[b] | K444S/V445K/G446V/Y449R/L452Q/N460K/K462R/ S477N/T478E/T478R/DEL483/E484P/F486I/F490P/ G496S/Q498Y/P499S/N501T/Y505H/V511I |
| SRR15202284 | NY | Biobot Analytics | 2021-06-28 | 98% | 16,281 | NY14 | ND[b] | K417T/K444S/DEL445-6/L452R/N460K/S477D/F486V/ Q493K/Q498Y/P499S/N501T |
| SRR15202279 | NY | Biobot Analytics | 2021-06-28 | 30% | 21,974 | NY11 | B.1 | N440K/K444S/DEL445-6/L452Q/Y453F/N460K/S477N/ D484/F486A/Q493K/Q498K/P499S/N501Y/H519N |
| SRR15128983 | NY | Biobot Analytics | 2021-06-16 | 99% | 21,152 | NY11 | A.29 | K444T/Y449H/E484A/Y489F/F490Y/Q498H |
| SRR15128978 | NY | Biobot Analytics | 2021-06-16 | 100% | 15,593 | NY10 | ND[b] | E484A/F486P/S494/Q498Y/H519N |

[a]n/a = not available; ND = none designated

the frequency of mutation A23056C (Q498H/Y), a mutation shared with the lineages in all 11 sewershed samples from NYC. We speculate that the relative abundance of cryptic lineages was high because, during this period, NYC experienced the lowest levels of COVID-19 infections seen since the start of the pandemic, and therefore the level of patient-derived SARS-CoV-2 RNA in wastewater was very low. As a result, the sequences that matched the known circulating lineage were at low abundance.

To compare the mutational profile among these different NYC samples, we first determined all of the mutations that occurred in at least 3 of the 11 cryptic lineages. We then produced a heat map to compare the frequency of each of these mutations from wastewater samples with the mutations that were reported from New York patient samples in June 2021 (Fig 8). Surprisingly, the sewershed sequences often lacked two of the four consensus sequences that define the B.1 PANGO lineage (GISAID G clades or Nextstrain '20' clades) of SARS-COV-2 [36]. Almost all patient samples collected in NYC during June 2020 contained the mutations C241T, C3037T, C14408T, and A23403G. The cryptic lineages from NYC wastewater all appeared to contain the mutations C3037T and A23403G, but possessed the ancestral

| Changes in genome | | | Patient Seqs | 15291304 | 15384049 | 15128978 | 15291305 | 15706711 | 16038156 | 15128983 | 15202279 | 16038150 | 15202285 | 15202284 |
|---|---|---|---|---|---|---|---|---|---|---|---|---|---|---|
| mutation | gene | AA change | NY, 6/22 | NY10 | NY10 | NY10 | NY11 | NY11 | NY11 | NY11 | NY11 | NY2 | NY2 | NY14 |
| C00241T | 5'UTR | - | 99% | 19% | 0% | | 0% | | 63% | 100% | | 63% | 0% | |
| C14,408T | Orf1b | P314L | 99% | 26% | 0% | 0% | 0% | 0% | 15% | 0% | 28% | 45% | | 0% |
| C3,037T | Orf1a | silent | 99% | 100% | 99% | | | | 100% | | | 100% | 100% | |
| A23,403G | S | D614G | 100% | 100% | 100% | 100% | 100% | 80% | 100% | 100% | 99% | 100% | | |
| C1,059T | Orf1a | T265I | 8% | 100% | 70% | 100% | | 91% | 67% | 100% | 48% | 100% | 0% | |
| A5,648C | Orf1a | K1795Q | 10% | 30% | 99% | | 100% | | 99% | 100% | 0% | 56% | | |
| A23,056C | S | Q498X | 0% | 100% | 99% | 100% | 100% | 100% | 92% | 99% | 0% | 79% | 0% | 98% |
| C24,044T | S | L828F | 0% | 72% | 99% | | | | 62% | 64% | 61% | 85% | | 0% |
| G25,563T | Orf3a | Q57H | 19% | 100% | 27% | | 53% | 70% | 81% | 100% | 99% | 79% | | 39% |
| C25,936G | Orf3a | H182D | 0% | 75% | 100% | 18% | 100% | | 70% | 100% | 30% | 83% | 0% | 100% |
| G25,947C | Orf3a | Q185H | 0% | 63% | 0% | 0% | 100% | | 75% | 24% | 11% | 70% | 0% | 0% |
| T27,322C | Orf6 | S41P | 0% | 42% | 66% | 100% | 0% | 90% | 48% | 0% | 0% | 41% | | 0% |
| C1,616A | Orf1a | L451I | 0% | 100% | 61% | 96% | 0% | 0% | 0% | 0% | | 0% | 100% | |
| C3,267T | Orf1a | T1001I | 32% | 100% | | 100% | | | 0% | | | 0% | | |
| G3,849T | Orf1a | S1195I | 0% | 81% | 99% | | 0% | 0% | 0% | | | 0% | | 0% |
| A4,178C | Orf1a | K1305Q | 0% | 60% | 100% | 0% | | 0% | 0% | 0% | 1% | 0% | | |
| C5,178A | Orf1a | T1638N | 0% | 87% | 53% | | 0% | | | 0% | 0% | 0% | 0% | |
| A6,328G | Orf1a | silent | 0% | 100% | 99% | 100% | | 0% | 0% | 0% | 0% | 0% | | 0% |
| T8,296C | Orf1a | silent | 0% | 98% | 100% | | 0% | | | 0% | 0% | 0% | | 0% |
| G22,599A | S | R346K | 10% | 100% | 72% | 51% | 0% | 0% | 0% | 0% | | 0% | 0% | |
| C23,039A | S | Q493K | 0% | 89% | 96% | 0% | 0% | 0% | 0% | 0% | 28% | 0% | 0% | 100% |
| C23,054T | S | Q498X | 0% | 100% | 99% | 100% | 0% | 0% | 0% | 1% | 0% | 0% | 100% | 99% |
| C23,117A | S | H519N | 0% | 100% | 99% | 99% | 0% | 0% | 0% | 1% | 53% | 0% | 0% | 0% |
| C23,277A | S | T572N | 0% | 100% | 99% | 52% | 0% | | 0% | 0% | 0% | 0% | | |
| T23,406C | S | V615A | 0% | 52% | 60% | 50% | 0% | 0% | 0% | 0% | 0% | 0% | | |
| G25,019A | S | D1153N | 0% | 75% | 99% | 100% | 0% | 0% | 0% | 0% | 0% | 0% | 0% | 0% |
| G25,116A | S | R1185H | 0% | 63% | 100% | 100% | 0% | | 0% | 0% | 0% | 0% | 0% | 52% |
| C28,887T | N | T205I | 12% | 89% | 77% | 71% | 0% | 0% | 0% | 0% | 0% | 9% | 100% | 54% |
| C4,113T | Orf1a | A1283V | 0% | 0% | 0% | 0% | 100% | 98% | 74% | 49% | 99% | 49% | | 0% |
| C4,230T | Orf1a | T1322I | 0% | 0% | 0% | 0% | | 96% | 73% | 50% | 99% | 0% | | |
| T5,507G | Orf1a | L1748V | 0% | 0% | | | 100% | | 89% | 100% | | 66% | | |
| A9,204G | Orf1a | D2980G | 0% | 41% | 0% | | 100% | | 81% | 100% | 0% | 100% | | 100% |
| G9,479T | Orf1a | G3072C | 0% | 0% | | | 0% | | 96% | 99% | 99% | 25% | 0% | |
| C9,711T | Orf1a | S3149F | 0% | 0% | 0% | 0% | 100% | 50% | 59% | 100% | 45% | 81% | 0% | 0% |
| T9,982C | Orf1a | silent | 0% | 0% | 0% | | 99% | | 50% | | 99% | 100% | 0% | |
| G11,670A | Orf1a | R3802H | 0% | 0% | 0% | 0% | 91% | 0% | 92% | | 94% | 76% | 0% | 0% |
| C11,916T | Orf1a | S3884L | 0% | 0% | 0% | 0% | 100% | 24% | 100% | 100% | 42% | 74% | 0% | 0% |
| G17,196A | Orf1b | silent | 0% | 0% | 0% | 0% | 100% | 70% | 82% | 100% | 65% | 56% | 0% | 0% |
| A17,496C | Orf1b | E1343D | 0% | 0% | 0% | | 100% | 89% | 35% | 100% | | 57% | 0% | |
| T18,660C | Orf1b | silent | 0% | 0% | 0% | 0% | 100% | 87% | 94% | 93% | 0% | 63% | 0% | 0% |
| G22,340A | S | A260T | 0% | 0% | | 0% | 93% | 84% | 100% | | | 41% | | 0% |
| A22,893C | S | K444T | 0% | 0% | 0% | 0% | 98% | 100% | 66% | 11% | 0% | 62% | 0% | 0% |
| T22,907C | S | Y449H | 0% | 0% | 0% | 0% | 99% | 100% | 86% | 99% | 0% | 57% | 100% | 0% |
| A23,013C | S | E484A | 0% | 9% | 3% | 100% | 100% | 100% | 99% | 99% | 0% | 75% | 100% | 0% |
| C23,029T | S | silent | 0% | 0% | 0% | 0% | 99% | 20% | 37% | 98% | 0% | 0% | 0% | 99% |
| T23,031A | S | F490Y | 0% | 0% | 0% | 0% | 99% | 100% | 100% | 98% | 0% | 79% | 4% | 0% |
| C24,418T | S | silent | 0% | 0% | 0% | 0% | 100% | | 72% | 100% | 0% | 63% | 0% | 0% |
| A25,020C | S | D1153A | 0% | 0% | 1% | 0% | 100% | 6% | 75% | 100% | 0% | 72% | 3% | 14% |
| T25,570A | Orf3a | S60T | 0% | 0% | 0% | | 0% | 71% | 54% | 56% | 1% | 0% | | 0% |
| A27,330C | Orf6 | silent | 0% | 0% | 0% | 0% | 100% | 94% | 74% | 99% | 99% | 66% | | 0% |
| T27,384C | Orf6 | silent | 1% | 0% | 0% | 0% | 100% | 35% | 41% | 100% | 68% | 58% | | 0% |
| T27,907G | Orf8 | V5G | 0% | 0% | 0% | | 100% | | 65% | | 99% | 100% | | 0% |
| C27,920T | Orf8 | silent | 0% | 15% | 0% | | 67% | | 27% | | 79% | 61% | | |
| T27,929A | Orf8 | silent | 0% | 0% | 0% | | 68% | | 28% | | 79% | 61% | | |
| A28,271T | UTR | - | 1% | 0% | 1% | 0% | 49% | | 47% | 0% | 27% | 56% | 0% | 0% |
| G29,540A | UTR | - | 0% | 0% | 0% | 0% | 99% | 84% | 71% | 88% | 78% | 65% | 0% | 0% |

**Fig 8. Polymorphisms from wastewater genomes.** Shown are all mutations present in at least three of the whole genome sequences from NYC listed in Table 2 and their corresponding amino acid changes. First column lists the prevalence of each mutation among all patients samples collected in June 2021 from New York. Each other column lists the prevalence of each mutation in each of the genome sequences.

sequences at positions 241 and 14408. In addition, there were two mutations in the S gene that were found in nearly all of the cryptic lineages, A23056C (Q498H/Y) and C24044T (L828F). Both of these mutations were found in less than 1% of patient samples. There were 3 additional mutations outside of the S gene that were highly prevalent in most of the wastewater samples, but essentially absent from patient samples: C25936G (Orf3 H182D), G25947C (Orf3 Q185H), and T27322C (Orf6 S41P). While other mutations were detected repeatedly within a sewershed, no other mutations spanned multiple sewersheds.

To confirm that some of the cryptic lineages lacked the B.1 lineage consensus mutations, we designed primers to amplify and sequence the C14408 region of SARS-CoV-2 RNA isolated from wastewater. Indeed, samples from NY11 and NY10 that had a high prevalence of cryptic lineages were found to contain sequences that lacked C14408T (S1 Fig). However, when samples were amplified from the NY13 sewershed when the cryptic lineages there were present, we observed only the modern C14408T, as would be expected if the NY13 lineage were derived from the Alpha VOC. In addition, we performed whole genome sequencing on a March 30, 2021 sample from MO33 when the cryptic lineages were highly prevalent and did not detect any sequence that lacked C241T or C14408T, suggesting the cryptic lineages in this sewershed diverged after the emergence of the B.1 lineage (S10 Data). Finally, we also analyzed the sequences from NCBI that contained the cryptic lineages from NJ and CA and did not find any sequences lacking C241T or C14408T. Thus, the lineages lacking C241T and C14408T appear to be limited to a subset of the cryptic lineages from NYC. These data are consistent with the hypothesis that a SARS-CoV-2 lineage bearing mutations C3037T and A23403G, but possessing the ancestral genotype at positions 241 and 14408, was the direct ancestor of most of the cryptic lineages found in NYC.

## 3. Discussion

Our results point to the evolution of numerous SARS-CoV-2 lineages under positive immune selection whose source/host remains unknown.

### 3.1 Relatedness of and origin of cryptic lineages

We previously detected cryptic lineages via targeted amplicon sequencing [8], but lacked information about their derivation. Here, from comparison of the sewersheds for which whole genome sequencing is available, it is clear that the cryptic lineages from wastewater are not all derived from a common ancestor. The NY13 lineage appeared to be derived from the Alpha VOC. If this is true, the NY13 lineage most likely branched off from Alpha sometime in early to mid-2021 when that variant was common in NYC. However, many lineages from the NY10, NY11, NY2, and NY14 sewersheds in New York appear to likely share a common ancestor that branched off from a pre-B.1 lineage. Additionally, we often observed swarms of related sequences that co-occurred within a sewershed on a single date, and accumulated new mutations over time, suggesting continued diversification from a single origin within each sewershed.

### 3.2 Comparison with the Omicron VOC

The Omicron VOC and the wastewater lineages appear to have been subjected to high positive selection. While prior VOCs had 3 or fewer amino acid changes in the amplified region of the

RBD, the Omicron VOC (BA.1) contained 11 and the cryptic lineages from wastewater averaged over 10. By comparison, a cluster of SARS-COV-2 sequences that appear to have circulated in white-tailed deer for over a year accumulated only 2 amino acid changes in this region [37]. Of the nonsynonymous RBD mutations in Omicron, four were in at least one prior VOC: K417N, S477N, T478K, and N501Y. The other seven were relatively rare; N440K was present in 0.2% of sequences and the other six were each present in less than 0.1% of sequences in GISAID [22–24] prior to November 1, 2021. All of the rare Omicron changes were observed in at least one of the cryptic wastewater lineages. Collectively, this suggests that the wastewater lineages and the Omicron VOC likely arose under similar selective pressures. The high dN/dS ratios found in cryptic lineages and in Omicron suggest that these selective pressures must be exceptionally strong.

## 3.3 Source of lineages

In spite of detailed tracking and cataloging of the cryptic lineages, the question where they are coming from remains unanswered. The most parsimonious explanations are 1) undetected spread within the human population, 2) prolonged shedding by individuals, or 3) spread in animal reservoirs.

Undetected spread in the population appears unlikely. While the sequencing rate for US patient samples is not 100%, it is high enough that population-level spread of cryptic lineages would not be missed. Alternatively, as it is known that SARS-CoV-2 can replicate in gastrointestinal sites [38,39], the lack of detection of cryptic lineages by clinical sequencing could be explained by the potential adaptation of some SARS-CoV-2 to replicate exclusively in the gastrointestinal tract [1,38]. Nonetheless, even if replication of these lineages were occurring outside of the nasopharyngeal region, this could not explain why cryptic lineages generally remain geographically constrained.

The most likely explanation for the appearance of cryptic lineages in wastewater is that they are shed by people with long-term COVID infections. Many such infections have been documented, particularly in immunosuppressed populations. Indeed, the vast majority of amino acid changes in the RBD of the Omicron VOC and the cryptic lineages confer resistance to neutralizing antibodies. In particular, substitutions at positions 417, 440, 460, 484, 493 and 501 have all been well documented to lead to immune evasion [17,27,34,40–42]. Additionally, RBD changes K417T, N440K, N460K, E484A, Q493K, and N501Y have all been observed in persistent infections of immunocompromised patients [43,44]. Given the repeated appearance of these mutations in diverse sewersheds, the majority of the selective pressure on the cryptic lineages is almost certainly immune pressure. A possible explanation for cryptic lineages is that they are the result of long-term SARS-CoV-2 infections of intestinal tissue. A recent paper reported extended presence of viral RNA in feces, long after it was undetectable in respiratory samples and suggested SARS-CoV-2 replication in the gastrointestinal (GI) tract could explain some of the symptoms associated to long-Covid [38]. The authors propose that SARS-CoV-2 infects the gastrointestinal tract and that some individuals shed the virus up to 7-months post-diagnosis.

The counterargument to cryptic lineages coming from patients is the sheer volume of viral shedding required to account for the wastewater signal. Many of the sewersheds process 50–100 million gallons of wastewater per day. Reliable amplification of a sequence from wastewater generally requires that the sequence is present at least 10,000 copies per liter. Therefore, detection of a specific virus lineage in such a sewershed would seem to require several trillion virus particles to be deposited each day. If this signal were derived from a single infected patient or even a small group of patients, those patients would have to shed exponentially more virus than typical COVID-19 patients.

The final explanation for the cryptic lineages in wastewater is that they are shed into wastewater by an animal host population. Previously, we determined through rRNA analysis of several NYC sewersheds that the major non-human mammals that contribute to the wastewater are cats, rats, and dogs [8]. Of these three, rats were the only species that seemed to be a plausible candidate. Indeed, we also showed that the cryptic lineages from the sewersheds had the ability to utilize rat and mouse ACE2 [8]. However, one of the sewersheds with the most consistent signal in 2021 was NY10, which had little to no rat rRNA. In addition, it is not clear why circulation in an immune competent animal such as a dog or a rat would result in a more rapid selection of immune escape mutations than circulation in humans, yet the cryptic lineages display accumulation of many times more immune escape changes than seen in viruses circulating in the human population.

### 3.4 The importance of wastewater sequencing methodology for identification of novel variants

To provide information regarding the appearance and spread of SARS-CoV-2 variants in communities, next generation sequencing technologies have been applied to sequence SARS-CoV-2 genetic material obtained from sewersheds around the world [45–47]. Commonly, SARS-CoV-2 RNA extracted from wastewater is amplified using SARS-CoV-2 specific primers that cover the entire genome [48–50]. Bioinformatic pipelines are employed to identify circulating SARS-CoV-2 variants [16,51]. In general, the presence and abundance of variants in wastewater corresponds to data obtained from clinical sequencing [45,46]. However, to our knowledge, there have been no other reports of cryptic lineages detected in wastewater that were not also observed in clinical sequence data. A major issue with generating whole genome sequence data from nucleic acid isolated from wastewater is sequence dropout over diagnostically important regions of the genome [48,52,53]. In some cases, diagnostically important regions of the genome that accumulate many mutations, such as the Spike RBD, receive little to no sequence coverage, making variant attribution difficult. Since wastewater contains a mixture of virus lineages and whole genome sequencing relies on sequencing of small genome fragments, mutations appearing on different reads cannot be linked together. Indeed, some variant identification pipelines map reads to reference genomes to estimate the probability that mutations are found in the same genome [16]. Such strategies would not be able to detect variants containing unique constellations of mutations. Detecting novel variants that are present at low relative abundances may be better achieved by targeted amplicon sequencing, such as the strategy we present here.

### 3.5 Summary

Over the past 15 months, cryptic SARS-CoV-2 lineages never seen in human patients have appeared in community wastewater in several locations across the USA [8]. These lineages have persisted, intermittently, often as swarms of closely related haplotypes that acquired additional amino acid changes over time, for up to 14 months. Evidence suggests that some of the lineages may have arisen during the initial phases of the pandemic in early to mid-2020. Significantly, these lineages often contained amino acid changes that have rarely or never appeared in contemporaneous variants, at least until the appearance of the Omicron VOC. Many of these amino acid changes are associated with evasion of antibody-mediated neutralization. Collectively, nonsynonymous substitutions in these lineages overwhelmingly outnumbered synonymous substitutions, indicating that these lineages have undergone exceptionally strong positive selection.

Three hypotheses for the origins of these lineages have been proposed: 1) undetected transmission, 2) long-term infections of immunocompromised patients and 3) possible animal reservoirs. Although immunosuppressed populations are the simplest explanation, it is difficult to reconcile the magnitude of the signal with individual patients being the source. Regardless of the origins and dynamics of cryptic variant shedding, our results highlight the ability of wastewater-based epidemiology to more completely monitor SARS-CoV-2 genetic diversity than can patient based sampling, at scale and at a greatly reduced cost. Given that multiple VOCs may have gone undetected until suddenly appearing, highly mutated, in apparently single evolutionary leaps [12], it is crucial to the early detection of the next variant of concern that novel SARS-CoV-2 genotypes are monitored for evidence of significant expansion. Importantly, patient sampling efforts, despite occurring with an intensity not seen in any prior epidemic, were unable to identify intermediary forms of many VOCs. Monitoring of wastewater, particularly using a targeted sequencing approach, likely provides the best avenue for detecting developing VOCs.

## 4. Materials and methods

### 4.1 Wastewater sample processing and RNA extraction

24-hr composite samples of wastewater were collected weekly from the inflow at each of the wastewater treatment plans.

NYC: Samples were processed on the day they were collected and RNA was isolated according to our previously published protocol [6]. Briefly, 250 mL from a 24-hr composite wastewater sample from each WWTP were centrifuged at 5,000 x g for 10 min at 4˚C to pellet solids. A 40 mL aliquot from the centrifuged samples was passed through a 0.22 μM filter (Millipore). To each corresponding filtrate, 0.9 g sodium chloride and 4.0 g PEG 8000 (Fisher Scientific) were added. The tubes were kept at 4˚C for 24 hrs and then centrifuged at 12,000 x g for 120 minutes at 4˚C to pellet the precipitate. The pellet was resuspended in 1.5 mL TRIzol (Fisher Scientific), and RNA was purified according to the manufacturer's instructions.

MO: Samples were processed as previously described [9]. Briefly, wastewater samples were centrifuged at 3000×g for 10 min and then filtered through a 0.22 μM polyethersulfone membrane (Millipore, Burlington, MA, USA). Approximately 37.5 mL of wastewater was mixed with 12.5 mL solution containing 50% (*w/vol*) polyethylene glycol 8000 and 1.2 M NaCl, mixed, and incubated at 4˚C for at least 1 h. Samples were then centrifuged at 12,000×g for 2 h at 4˚C. Supernatant was decanted and RNA was extracted from the remaining pellet (usually not visible) with the QIAamp Viral RNA Mini Kit (Qiagen, Germantown, MD, USA) using the manufacturer's instructions. RNA was extracted in a final volume of 60 μL.

CA: Samples were processed as previously described [54]. Briefly, 40 mLs of influent was mixed with 9.35g NaCl and 400 uL of 1M Tris pH 7.2, 100mM EDTA. Solution was filtered through a 5-um PVDF filter and 40 mLs of 70% EtNY11 was added. Mixture was passed through a silica spin column. Columns were washed with 5 mL of wash buffer 1 (1.5 M NaCl, 10 mM Tris pH 7.2, 20% EtNY11), and then 10 mL of wash buffer 2 (100 mM NaCl, 10 mM Tris pH 7.2, 80% EtNY11). RNA was eluted with 200 ul of ZymoPURE elution buffer.

### 4.2 Targeted PCR: MiSeq sequencing

The primary RBD RT-PCR was performed using the Superscript IV One-Step RT-PCR System (Thermo Fisher Scientific,12594100). Primary RT-PCR amplification was performed as follows: 25˚C (2:00) + 50˚C (20:00) + 95˚C (2:00) + [95˚C (0:15) + 55˚C (0:30) + 72˚C (1:00)] × 25 cycles using the MiSeq primary PCR primers CTGCTTTACTAATGTCTATGCAGATTC and NCCTGATAAAGAACAGCAACCT. Secondary PCR (25 μL) was performed on RBD

amplifications using 5 μL of the primary PCR as template with MiSeq nested gene specific primers containing 5′ adapter sequences (0.5 μM each) acactctttccctacacgacgctcttccgatctG-TRATGAAGTCAGMCAAATYGC and gtgactggagttcagacgtgtgctcttccgatctATGTCAA-GAATCTCAAGTGTCTG, dNTPs (100 μM each) (New England Biolabs, N0447L) and Q5 DNA polymerase (New England Biolabs, M0541S). Secondary PCR amplification was performed as follows: 95˚C (2:00) + [95˚C (0:15) + 55˚C (0:30) + 72˚C (1:00)] × 20 cycles. A tertiary PCR (50 μL) was performed to add adapter sequences required for Illumina cluster generation with forward and reverse primers (0.2 μM each), dNTPs (200 μM each) (New England Biolabs, N0447L) and Phusion High-Fidelity or (KAPA HiFi for CA samples) DNA Polymerase (1U) (New England Biolabs, M0530L). PCR amplification was performed as follows: 98˚C (3:00) + [98˚C (0:15) + 50˚C (0:30) + 72˚C (0:30)] × 7 cycles +72˚C (7:00). Amplified product (10 μl) from each PCR reaction is combined and thoroughly mixed to make a single pool. Pooled amplicons were purified by addition of Axygen AxyPrep MagPCR Clean-up beads (Axygen, MAG-PCR-CL-50) or in a 1.0 ratio to purify final amplicons. The final amplicon library pool was evaluated using the Agilent Fragment Analyzer automated electrophoresis system, quantified using the Qubit HS dsDNA assay (Invitrogen), and diluted according to Illumina's standard protocol. The Illumina MiSeq instrument was used to generate paired-end 300 base pair reads. Adapter sequences were trimmed from output sequences using Cutadapt.

## 4.3 Long PCR and subcloning

The long RBD RT-PCR was performed using the Superscript IV One-Step RT-PCR System (Thermo Fisher Scientific, 12594100). Primary long RT-PCR amplification was performed as follows: 25˚C (2:00) + 50˚C (20:00) + 95˚C (2:00) + [95˚C (0:15) + 55˚C (0:30) + 72˚C (1:30)] × 25 cycles using primary primers CCCTGCATACACTAATTCTTTCAC and TCCTGA-TAAAGAACAGCAACCT. Secondary PCR (25 μL) was performed on RBD amplifications using 5 μL of the primary PCR as template with nested primers (0.5 μM each) CATTCAACT-CAGGACTTGTTCTT and ATGTCAAGAATCTCAAGTGTCTG, dNTPs (100 μM each) (New England Biolabs, N0447L) and Q5 High-Fidelity DNA Polymerase (New England Biolabs, M0491L). Secondary PCR amplification was performed as follows: 95˚C (2:00) + [95˚C (0:15) + 55˚C (0:30) + 72˚C (1:30)] × 20 cycles.

Positive amplifications were visualized in an agarose gel stained with ethidium bromide, excised, and purified with a NucleoSpin Gel and PCR Clean-up Kit (Macherey-Nagel, 74609.250). Gel purified DNA was subcloned using a Zero Blunt TOPO PCR Cloning Kit (Invitrogen, K2800-20SC). Individual colonies were transferred to capped test tubes containing 10 ml of 2X YT broth (ThermoFisher, BP9743-5). Test tubes were incubated at 37˚C and shook at 250 rpm for 24 hours. The resulting *E. Coli* colonies were centrifuged for 10 minutes at 5000 xg and the supernatant was decanted. Plasmid DNA was extracted from the pellet using a GeneJet Plasmid Miniprep Kit (ThermoFisher, K0503). The concentration of plasmid DNA extracts was measured using a NanoDrop One (ThermoFisher, ND-ONE-W).

## 4.4 PacBio sequencing

A nested RT-PCR protocol was used to generate 1.6kb Spike amplicons from wastewater RNAs for PacBio sequencing. The primary RT-PCR amplification was performed with the Superscript IV One-Step RT-PCR System (Invitrogen) and the same thermal cycling program as described above for MiSeq amplicons. These inter Spike gene-specific primer sequences (5'-[*BC10ab*]-ATTCAACTCAGGACTTGTTCTT and 5'-[*BC10xy*]-ATGTCAAGAATCT-CAAGTGTCTG) were tagged directly on their 5' ends with standard 16 bp PacBio barcode

sequences and used with asymmetric barcode combinations that allow large numbers of samples to be pooled prior to sequencing. The following thermal cycling profile was used for nested PCR: 98˚C (2 min) + [98˚C (10 sec) + 55˚C(10 sec) + 72˚C (1 min)] x 20 cycles + 72˚C (5 min). The resulting PCR amplicons were then subjected to three rounds of purification with AMPure XP beads (Beckman Coulter Life Sciences) in a ratio of 0.7:1 beads to PCR. Purified amplicons were quantified using a Qubit dsDNA HS kit (ThermoFisher Scientific) and pooled prior to PacBio library preparation.

After ligation of SMRTbell adaptors according to the manufacturer's protocol, sequencing was completed on a PacBio Sequel II instrument (PacBio, Menlo Park, CA USA) in the Genomic Sequencing Laboratory at the Centers of Disease Control in Atlanta, GA, USA. Raw sequence data was processed using the SMRT Link v10.2 command line toolset (Software downloads - PacBio). Circular consensus sequences were demultiplexed based on the asymmetric barcode combinations and subjected to PB Amplicon Analysis to obtain high-quality consensus sequences and search for minor sequence variants.

## 4.5 Bioinformatics

**4.5.1 MiSeq and PacBio processing.** Sequencing reads were processed as previously described. Briefly, VSEARCH tools were used to merge paired reads and dereplicate sequences [55]. Dereplicated sequences from RBD amplicons were mapped to the reference sequence of SARS-CoV-2 (NC_045512.2) spike ORF using Minimap2 [56]. Mapped amplicon sequences were then processed with SAM Refiner using the same spike sequence as a reference and the command line parameters "—Alpha 1.8—foldab 0.6" [9].

The covariant deconvolution outputs were used to generate the haplotype plots in Figs 1–7. Covar outputs of SAM Refiner for MiSeq sequences were collected by sewershed and multiple runs of the same sample averaged. The collected sequence data were processed to determine core haplotypes of cryptic lineages observed in each sewershed. First sequences that contained only one or no variation relative to the reference Wuhan I sequence were discarded. Remaining sequences with 6 or fewer variations and containing the polymorphisms defining Alpha, Beta, Gamma or Delta were assigned to the defining haplotype of the matching VOC. Any sequences not reassigned with fewer than 4 variations were removed. Sequences with at least 6 variations were matched against Omicron BA.1, BA.2 and BA.5. Sequences that matched an Omicron lineage with more than 70% identity were assigned to the defining haplotype for the matching lineage. Remaining unassigned sequences were then processed to remove polymorphisms that did not appear in at least two sample dates (except for MO45 and California sequences, due to the small number of samples with cryptic sequences) or never appeared in a sample at an abundance greater than .5%. In-frame deletions bypassed this removal. Condensed sequences that appear in at least two samples or had a summed abundance of at least 2% across all samples were passed on to further steps. The above process was reiterated until no more processing occurred. Non-VOC sequences were then aligned via MAFFT [57] and then all sequences rendered into figures using plotnine https://plotnine.readthedocs.io/en/stable/index.html. The PacBio sequences were similarly collected to generate the haplotype plot in Fig 7, without the polymorphism condensation or alignment.

**4.5.2 NCBI SRA screening.** Raw reads were downloaded and then processed similar to MiSeq sequencing except the reads were mapped to the entire SARS-CoV-2 genome and SAM Refiner was run with the parameters '—wgs 1—collect 0—indel 0—covar 0—min_count 1—min_samp_abund 0—min_col_abund 0—ntabund 0—ntcover 1'. Unique sequence outputs from SAM Refiner were then screened for specific amino acid changes. The nt call outputs of samples of interest were used to determine other variations in the genomes sequenced.

**4.5.3 14408 sequencing.** The long RBD RT-PCR was performed using the Superscript IV One-Step RT-PCR System (Thermo Fisher Scientific, 12594100). Primary long RT-PCR amplification was performed as follows: 25˚C (2:00) + 50˚C (20:00) + 95˚C (2:00) + [95˚C (0:15) + 55˚C (0:30) + 72˚C (1:30)] × 25 cycles using primary primers ATACAAACCACGC-CAGGTAG and AACCCTTAGCACACAGCAAAGT. Secondary PCR (25 μL) was performed on RBD amplifications using 5 μL of the primary PCR as template with nested primers (0.5 μM each) ACACTCTTTCCCTACACGACGCTCTTCCGATCTGGTAGTG-GAGTTCCTGTTGTAG and GTGACTGGAGTTCAGACGTGTGCTCTTCCGATCTAG-CACGTAGTGCGTTTATCT, dNTPs (100 μM each) (New England Biolabs, N0447L) and Q5 High-Fidelity DNA Polymerase (New England Biolabs, M0491L). Secondary PCR amplification was performed as follows: 95˚C (2:00) + [95˚C (0:15) + 55˚C (0:30) + 72˚C (1:30)] × 20 cycles.

**4.5.2 Whole genome sequencing.** Whole genome sequencing of the SARS-CoV-2 genome from the MO33 sewershed was performed using the NEBNext ARTIC SARS-CoV-2 Library Prep Kit (Illumina). Amplicons were sequenced on an Illumina MiSeq instrument. Output sequences were analyzed using the program SAM Refiner [58].

## Supporting information

**S1 Fig. Sequence of nt 14408 from NYC wastewater.**
(TIFF)

**S1 Table. Prevalence in GISAID of common substitutions found in cryptic lineages.**
(DOCX)

**S1 Data. Dereplicated MO33 sequences from RBD amplicons were mapped to the reference sequence of SARS-CoV-2 (NC_045512.2) spike ORF using Minimap.** Mapped amplicon sequences were then processed with SAM Refiner using the same spike sequence as a reference and the command line parameters "—Alpha 1.8—foldab 0.6".
(TSV)

**S2 Data. Dereplicated MO45 sequences from RBD amplicons were mapped to the reference sequence of SARS-CoV-2 (NC_045512.2) spike ORF using Minimap.** Mapped amplicon sequences were then processed with SAM Refiner using the same spike sequence as a reference and the command line parameters "–Alpha 1.8 –foldab 0.6".
(TSV)

**S3 Data. Dereplicated NY3 sequences from RBD amplicons were mapped to the reference sequence of SARS-CoV-2 (NC_045512.2) spike ORF using Minimap.** Mapped amplicon sequences were then processed with SAM Refiner using the same spike sequence as a reference and the command line parameters "—Alpha 1.8—foldab 0.6".
(TSV)

**S4 Data. Dereplicated NY14 sequences from RBD amplicons were mapped to the reference sequence of SARS-CoV-2 (NC_045512.2) spike ORF using Minimap.** Mapped amplicon sequences were then processed with SAM Refiner using the same spike sequence as a reference and the command line parameters "—Alpha 1.8—foldab 0.6".
(TSV)

**S5 Data. Dereplicated NY10 sequences from RBD amplicons were mapped to the reference sequence of SARS-CoV-2 (NC_045512.2) spike ORF using Minimap.** Mapped amplicon sequences were then processed with SAM Refiner using the same spike sequence as a reference

and the command line parameters "—Alpha 1.8—foldab 0.6".
(TSV)

**S6 Data. Dereplicated NY11 sequences from RBD amplicons were mapped to the reference sequence of SARS-CoV-2 (NC_045512.2) spike ORF using Minimap.** Mapped amplicon sequences were then processed with SAM Refiner using the same spike sequence as a reference and the command line parameters "—Alpha 1.8—foldab 0.6".
(TSV)

**S7 Data. Dereplicated NY2 sequences from RBD amplicons were mapped to the reference sequence of SARS-CoV-2 (NC_045512.2) spike ORF using Minimap.** Mapped amplicon sequences were then processed with SAM Refiner using the same spike sequence as a reference and the command line parameters "—Alpha 1.8—foldab 0.6".
(TSV)

**S8 Data. Dereplicated NY13 sequences from RBD amplicons were mapped to the reference sequence of SARS-CoV-2 (NC_045512.2) spike ORF using Minimap.** Mapped amplicon sequences were then processed with SAM Refiner using the same spike sequence as a reference and the command line parameters "—Alpha 1.8—foldab 0.6".
(TSV)

**S9 Data. Dereplicated CA sequences from RBD amplicons were mapped to the reference sequence of SARS-CoV-2 (NC_045512.2) spike ORF using Minimap.** Mapped amplicon sequences were then processed with SAM Refiner using the same spike sequence as a reference and the command line parameters "—Alpha 1.8—foldab 0.6".
(TSV)

**S10 Data. Whole genome sequencing of the SARS-CoV-2 genome from the MO33 sewershed.** Shown is the nt_calls output from SAMRefiner.
(TSV)

## Acknowledgments

The authors thank Benjamin Martin-Rambo, Dhwani Batra, Kristine Lacek, Sarah Nobles, and Justin Lee at the Centers for Disease Control and Prevention Genomic Sequencing Lab for assistance with PacBio sequencing. We also thank Thomas Peacock for valuable advice and feedback during the preparation of this manuscript. Thanks to Kristen Cheung, Anna Gao, Nanami Kubota, and Shyanon Rai for experimental assistance.

## Author Contributions

**Conceptualization:** Devon A. Gregory, Rose S. Kantor, John J. Dennehy, Marc C. Johnson.

**Data curation:** Devon A. Gregory, Monica Trujillo, Clayton Rushford, Anna Flury, Sherin Kannoly, Kaung Myat San, Dustin T. Lyfoung, Roger W. Wiseman, Karen Bromert, Ming-Yi Zhou, Ellen Kesler, Nathan J. Bivens, Jay Hoskins, Chung-Ho Lin, David H. O'Connor, Rose S. Kantor, John J. Dennehy, Marc C. Johnson.

**Formal analysis:** Devon A. Gregory, David H. O'Connor, Rose S. Kantor, John J. Dennehy, Marc C. Johnson.

**Funding acquisition:** David H. O'Connor, Jeff Wenzel, Rose S. Kantor, John J. Dennehy, Marc C. Johnson.

**Investigation:** Devon A. Gregory, Monica Trujillo, Clayton Rushford, Anna Flury, Sherin Kannoly, Kaung Myat San, Dustin T. Lyfoung, Roger W. Wiseman, Karen Bromert, Ming-Yi Zhou, Ellen Kesler, Nathan J. Bivens, Jay Hoskins, Chung-Ho Lin, David H. O'Connor, Rose S. Kantor, John J. Dennehy, Marc C. Johnson.

**Methodology:** Devon A. Gregory, David H. O'Connor, John J. Dennehy, Marc C. Johnson.

**Project administration:** Nathan J. Bivens, David H. O'Connor, Chris Wieberg, Jeff Wenzel, Rose S. Kantor, John J. Dennehy, Marc C. Johnson.

**Resources:** David H. O'Connor, Rose S. Kantor, John J. Dennehy, Marc C. Johnson.

**Software:** Devon A. Gregory, David H. O'Connor, Rose S. Kantor.

**Supervision:** Nathan J. Bivens, David H. O'Connor, Jeff Wenzel, Rose S. Kantor, John J. Dennehy, Marc C. Johnson.

**Visualization:** Devon A. Gregory, Rose S. Kantor, John J. Dennehy, Marc C. Johnson.

**Writing – original draft:** Devon A. Gregory, Rose S. Kantor, John J. Dennehy, Marc C. Johnson.

**Writing – review & editing:** Devon A. Gregory, Monica Trujillo, David H. O'Connor, Chris Wieberg, Jeff Wenzel, Rose S. Kantor, John J. Dennehy, Marc C. Johnson.

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
