## [Decision Letter · Decision Letter 0]

21 Jul 2022

Dear Dr. Johnson,

Thank you very much for submitting your manuscript "Genetic Diversity and Evolutionary Convergence of Cryptic SARS-CoV-2 Lineages Detected Via Wastewater Sequencing" for consideration at PLOS Pathogens. As with all papers reviewed by the journal, your manuscript was reviewed by members of the editorial board and by several independent reviewers. In light of the reviews (below this email), we would like to invite the resubmission of a significantly-revised version that takes into account the reviewers' comments.

I have secured three expert reviews of your study. Although the topic is considered to be of high interest, all three reviewers concerns raised about novelty; specifically, whether the manuscript constitutes a sufficiently significant advance beyond your team's recently published work on this subject (Smyth et al. 2022, PMID: 35115523). Additionally, a number of issues were raised concerning technical rigor: Reviewer 1 took issue with the PCR approach used, raising a concern about the potential for PCR chimerism and the potential for bias related to so-called "jackpot amplification." All three reviewers raised concerns with the computational and bioinformatic analyses on which your conclusions were based. Reviewer 2 flagged a lack of clarity about the analytical methods used to resolve multiple SARS-CoV-2 lineages from mixed samples, echoing the methodological concern brought up by Reviewer 1. Reviewer 3 provided useful feedback about how your results are presented, and requested more clarity on how well S-genotype based lineage assignments compare to assignments made using whole genome data. Given the extensive evidence for convergent evolution in Spike, this is a valid question. The reviewers provided a number of specific suggestions for improving the study in a substantial (major) revision. I would therefore like to provide you and your colleagues the opportunity to address their concerns. Please pay particular attention to the feedback concerning novelty, as this was a key point raised by all of the reviewers.

We cannot make any decision about publication until we have seen the revised manuscript and your response to the reviewers' comments. Your revised manuscript is also likely to be sent to reviewers for further evaluation.

Sincerely,

Jeremy P. Kamil, Ph.D.

Guest Editor

PLOS Pathogens

Marco Vignuzzi

Section Editor

PLOS Pathogens

Kasturi Haldar

Editor-in-Chief

PLOS Pathogens

orcid.org/0000-0001-5065-158X

Michael Malim

Editor-in-Chief

PLOS Pathogens

orcid.org/0000-0002-7699-2064

Dear Dr. Johnson and co-authors,

Thank you for submitting your manuscript to PLoS Pathogens. I have secured three expert reviews of your study. Although the topic is considered to be of high interest, all three reviewers concerns raised about novelty; specifically, whether the manuscript constitutes a sufficiently significant advance beyond your team's recently published work on this subject (Smyth et al. 2022, PMID: 35115523). Additionally, a number of issues were raised concerning technical rigor: Reviewer 1 took issue with the PCR approach used, raising a concern about the potential for PCR chimerism and the potential for bias related to so-called "jackpot amplification." All three reviewers raised concerns with the computational and bioinformatic analyses on which your conclusions were based. Reviewer 2 flagged a lack of clarity about the analytical methods used to resolve multiple SARS-CoV-2 lineages from mixed samples, echoing the methodological concern brought up by Reviewer 1. Reviewer 3 provided useful feedback about how your results are presented, and requested more clarity on how well S-genotype based lineage assignments compare to assignments made using whole genome data. Given the extensive evidence for convergent evolution in Spike, this is a valid question. The reviewers provided a number of specific suggestions for improving the study in a substantial (major) revision. I would therefore like to provide you and your colleagues the opportunity to address their concerns. Please pay particular attention to the feedback concerning novelty, as this was a key point raised by all of the reviewers.

Reviewer's Responses to Questions

**Part I - Summary**

Reviewer #1: This is an extremely interesting paper describing the identification of highly mutation RBD sequences in SARS-CoV-2 wastewater sequencing. Of note, many of the mutations are key antigenic changes, suggesting they may have arisen under immune selection and to some degree could foreshadow the subsequent evolution of variants.

The exact source of these sequences is unclear. The authors provide a reasonable discussion of several hypotheses, none of which seem entirely satisfactory although I agree chronic shedding is probably the most likely. However, I don't have a better hypothesis, and I think this is in the end a paper that will mostly describe the empirical result without a completely satisfying explanation. But given the importance of the result, this should certainly be studied to provoke continued study.

I do have some major comments, below. These are mostly related to ensuring the robustness of the results to some potential technical and computational analysis issues.

Reviewer #2: In this research article, Gregory et al. report the unique detection of partial SARS-CoV-2 Spike sequences in 9 sewersheds in the U.S. termed here as “cryptic lineages”. This same group previously reported detection of such unusual sequences in three sewersheds in NYC earlier this year (Smyth et al. 2022 Nat. Comm.). These data add some additional depth to the prior report by demonstrating that detection of these lineages is not limited to a single geographical location (now includes Missouri, California, and NYC). Most of the figures comprise of representations of singular sewershed sequence distributions with little direct comparison between them. Additional phylogenetic analyses would be required to support some of the observations. Ultimately, little additional insight is gained as to the source of these sequences with the same potential explanations provided as in the prior report. Substantial revisions to the analyses performed and data representation are advised to differentiate and/or highlight new conclusions from this work.

Reviewer #3: The manuscript "Genetic Diversity and Evolutionary Convergence of Cryptic SARS-CoV-2 Lineages Detected Via Wastewater Sequencing" by Gregory et al. describes an approach to amplify and sequence the receptor-binding domain (RBD) of SARS-COV-2, aiding researchers in characterizing viral lineages present in wastewater. This is not entirely novel since the same authors have other similar manuscripts. The employed methodology allowed the researchers to monitor SARS-CoV-2 lineages circulating for months in different areas, which may be helpful. Authors claimed that their findings demonstrate that "SARS-COV-2 genetic diversity is greater than what is commonly observed through routine SARS-CoV-2 surveillance", which was already reported in other manuscripts. The authors speculate that one of the possible sources of the cryptic lineages observed in sewage, but never found in human cases, may have their origin from animal reservoirs. However, there is no substantial evidence to support this hypothesis.

**Part II – Major Issues: Key Experiments Required for Acceptance**

Reviewer #1: Have any controls being performed to make sure the sequencing is actually capturing viral RNA rather than DNA? Both New York City and parts of Missouri are home to major research universities that are performing experiments on SARS-CoV-2, including construction of plasmids encoding mutants of the SARS-CoV-2 spike. For instance, in New York City the Bieniasz lab at Rockefeller created a polymutant spike in VSV that has many RBD mutations. Presumably some of the DNA waste from these experiments also ends up going down the drain. I know this seems unlikely to explain all the results, but some controls (e.g., no-reverse-transcriptase preps of samples that contain RBD) should be performed just to ensure the sequencing isn't capturing DNA from experiments.

Many PCR cycles (~50) are performed to amplify the RBD for sequencing. This creates two potential concerns: PCR chimerism (where different mutations from different molecules are scrambled due to recombination during the PCR reaction) and jackpot amplification of specific molecules. For the jackpot amplification, this can be largely corrected for by splitting the original sample into halves, performing the entire prep and sequencing separately, and then seeing if the same mutants are identified. It is somewhat unclear how systematically this was done: the methods say something about "sequences that appeared in only one sample... were discarded", but more elaboration on this point would be useful. For the PCR chimerism (recombination), it would be useful to mention this more prominently as a caveat particularly with respect to different sequences that have subsets of the same pool of mutations.

Overall, more description of the exact cutoffs used to call the variants, and the sensitivity of the results to these cutoffs, should be included in the main text Results.

The analysis focuses almost entirely on the amino-acid mutations, although it tangentially mentions that the number of synonymous mutations is low. This analysis should be elaborated, including formal analysis of whether dN/dS exceeds 1 (the number of amino acid mutations exceeds the number of synonymous mutations more than expected for random nucleotide mutations).

Reviewer #2: 1) While the author’s use of the term ‘lineage’ is defined at the beginning of the results section and is useful for messaging, it is unclear how S-genotype based lineage designations do or don’t compare to whole-genome sequencing based lineage or clade designations. For example, is clade 20A different from 20G in this region? Or Alpha and Beta? It would be useful to have a SARS-CoV-2 phylogenetic tree built from just the sequenced region to illustrate the variants that can or cannot be cleanly differentiated by this method.

2) It is somewhat unclear how much diversity in these samples is detected that is attributed to clinically-derived specimens and known lineages. It would be particularly informative to generate a phylogenetic tree that visualizes all S-genotype diversity in a given wastewater specimen compared to a time-matched tree of the same region of S from clinical specimens sampled over a similar time frame. In theory, they should look very similar with the exception of the ‘cryptic lineages’ and doing such would provide statistical support for the existence of the unique lineage in the specimen. This wouldn’t need to be done exhaustively, but would add confidence to the analysis.

3) The representations used for the S-genotypes in Figures 1-6 are redundant in showing results from assorted sewersheds and do not easily reflect the main points raised with each of the figures. Ideally, the sequences would be grouped on the y-axis by phylogenetic relatedness as opposed to by time. This could also provide statistical support for the somewhat arbitrarily defined ‘classes’ in Figure 4/5. The reference sequence should be more readily apparent at the top and the addition of common comparator sequences of VOCs could be added to make some points more salient (as done in Figure 7). Phylodynamic models of statistically supported lineage groups over time would be better, which could even be annotated with weighted nodes to reflect frequency as has been done for representing intra-host diversity. If this is not done, some claims about viral origins and evolution will need to be removed or discussed as hypothetical as they would lack statistical support.

Reviewer #3: It is unclear to this reviewer how the authors handled resolving multiple SARS-CoV-2 lineages from mixed samples, which is harder using short-read sequence data. For example, please see the Freyja approach in https://www.ncbi.nlm.nih.gov/pmc/articles/PMC8996633/ and detail your strategy to overcome this issue. This is a crucial point for all studies using environmental samples.

Page 32, lines 500-503. Although waste-water surveillance may contribute to SARS-CoV-2 epidemiology studies, the results described in the present manuscript may also show that some mutation constellations found in the environment apparently do not circulate between humans. The reason for that is still lacking. Therefore, it is hard to accept the statement “our results highlight the ability of wastewater-based epidemiology to more completely monitor SARS-CoV-2 transmission and genetic diversity than can patient based sampling”.

Methods section: Three different protocols were used for water concentration and RNA extraction. Do the authors have a comparison on that? I mean, one of those had a better performance to recover viral RNA? How have these different protocols affected genome recovery and assembly?

**Part III – Minor Issues: Editorial and Data Presentation Modifications**

Reviewer #1: (No Response)

Reviewer #2: 1) While the authors report 9 sewersheds with cryptic lineages, it is unclear how many they performed surveillance on. In other words, it should be reported how many systems and specimens had no aberrant sequences.

2) It is really interesting to note that some of the cryptic lineages monitored showed stability in the sewershed over a period of time where clinical specimens varied wildly. For example, NY3, which had a lineage that persisted through the major Alpha, Delta, and then Omicron shifts.

3) The claims of where cryptic lineages derived from can really only be made in the context of a true phylogenetic comparison (i.e., lines 190-195, 217-219, etc).

4) A direct comparison of cryptic lineages would add substantial value. Some ad hoc comparisons are provided (i.e., lines 201-207), but it is really hard to try to compare them using the given representations.

5) Update the Omicron lineage comparison in lines 213-217 to account for new mutations in the sequenced S-gene region in sublineages BA.2-BA.5.

6) Reservoirs of SARS-CoV-2 in the gut have been suggested to be a contributor to long COVID. This may want to be added to the Discussion.

Reviewer #3: How wastewater positivity correlates (spatially) with positive clinical cases in the present study?

It is unclear to this reviewer why the authors emphasize that “Many of the amino acid substitutions in these lineages occurred at residues also mutated in the Omicron variant of concern (VOC), often with the same substitution.” Do they suggest any relationship with Omicron emergence? Please clarify.

Do the authors have an explanation, or hypotheses, for cryptic lineages lasting for around six months without detection on human samples?

Figure 1. Some of those positions that authors call “Omicron position” were detected before in other VOCs like Alpha, Beta, and Gamma. Thus, this reviewer suggests that the “Omicron position” is not the best option.

Page 8, lines 151-152. Authors stated, “Amino acid changes similar to (green boxes) or identical to (orange boxes) changes in Omicron (BA.1) are indicated.” In fact, to Omicron VOC, but not all observed to the BA.1 lineage. Please see https://outbreak.info/situation-reports?pango=BA.1

Page 19, lines 300-302. Authors stated: “This second sequence presumably represented a unique lineage that had not been detected by routine surveillance.” Since the USA has extensive genomics surveillance, authors should comment on the clinical surveillance specific in this area.

Page 19, lines 303-305. This elegant result further confirmed the VOC Alpha-related origin of the NY13 lineage.

Figure 7. I would suggest including the residue positions and the nucleotide positions according to the Wuhan RefSeq.

Figure 7. How many samples underwent each sequencing protocol (e.g., Sanger or PacBio)? How many samples were analyzed for minor variants in the whole study? Was it enough for statistical analysis?

Page 21, lines 341-343. Authors stated: “We speculate that the relative abundance of cryptic lineages was high because, during this period, NYC experienced the lowest levels of COVID-19 infections seen since the start of the pandemic.” Please, clarify this statement.

Page 31, lines 480-482. What about probe-based genome capture? These protocols can generate complete genomes even with high Ct levels. Authors should comment on this.

Page 32, lines 506-508. Authors stated: “Importantly, patient sampling efforts, despite occurring with an intensity not seen in any prior epidemic, were unable to identify intermediary forms of most VOCs”.

That is not necessarily true. See the manuscript by Gräf and colleagues (https://academic.oup.com/ve/article/7/2/veab091/6462077) about the origins of VOC Gamma.

Another critical point is that we now have much more data about mutations of concern that may aid in detecting a new VOC.

PLOS authors have the option to publish the peer review history of their article (what does this mean?). If published, this will include your full peer review and any attached files.

Reviewer #1: No

Reviewer #2: No

Reviewer #3: No
---

## [Editor Report · Decision Letter 1]

22 Sep 2022

Dear Dr. Johnson,

We are pleased to inform you that your manuscript 'Genetic Diversity and Evolutionary Convergence of Cryptic SARS-CoV-2 Lineages Detected Via Wastewater Sequencing' has been provisionally accepted for publication in PLOS Pathogens.

Best regards,

Jeremy P. Kamil, Ph.D.

Guest Editor

PLOS Pathogens

Marco Vignuzzi

Section Editor

PLOS Pathogens

Kasturi Haldar

Editor-in-Chief

PLOS Pathogens

orcid.org/0000-0001-5065-158X

Michael Malim

Editor-in-Chief

PLOS Pathogens

orcid.org/0000-0002-7699-2064

Dear Dr. Johnson and colleagues,

Thank you for addressing the concerns of the reviewers and sending in this revision of your manuscript. I am recommending that the Editorial Board accept your study for publication.
---

## [Editor Report · Acceptance letter]

6 Oct 2022

Dear Dr. Johnson,

We are delighted to inform you that your manuscript, "Genetic Diversity and Evolutionary Convergence of Cryptic SARS-CoV-2 Lineages Detected Via Wastewater Sequencing," has been formally accepted for publication in PLOS Pathogens.

Best regards,

Kasturi Haldar

Editor-in-Chief

PLOS Pathogens

orcid.org/0000-0001-5065-158X

Michael Malim

Editor-in-Chief

PLOS Pathogens

orcid.org/0000-0002-7699-2064